# Behind the Machine's Gaze: Neural Networks with Biologically-inspired Constraints Exhibit Human-like Visual Attention

**Leo Schwinn**[1,2]                                                    *leo.schwinn@fau.de*
**Doina Precup**[2,3,4]                                                 *dprecup@cs.mcgill.ca*
**Bjoern M. Eskofier**[1]                                              *bjoern.eskofier@fau.de*
**Dario Zanca**[1]                                          *dario.zanca@fau.de (corr. author)*

[1] *Friedrich-Alexander-Universität Erlangen-Nürnberg*     [2] *Mila*     [3] *McGill University*     [4] *DeepMind*

**Reviewed on OpenReview:** *https://openreview.net/forum?id=7iSYW1FRWA*

## Abstract

By and large, existing computational models of visual attention tacitly assume perfect vision and full access to the stimulus and thereby deviate from foveated biological vision. Moreover, modeling top-down attention is generally reduced to the integration of semantic features without incorporating the signal of a high-level visual tasks that have been shown to partially guide human attention. We propose the Neural Visual Attention (NeVA) algorithm to generate visual scanpaths in a top-down manner. With our method, we explore the ability of neural networks on which we impose a biologically-inspired foveated vision constraint to generate human-like scanpaths without directly training for this objective. The loss of a neural network performing a downstream visual task (i.e., classification or reconstruction) flexibly provides top-down guidance to the scanpath. Extensive experiments show that our method outperforms state-of-the-art unsupervised human attention models in terms of similarity to human scanpaths. Additionally, the flexibility of the framework allows to quantitatively investigate the role of different tasks in the generated visual behaviors. Finally, we demonstrate the superiority of the approach in a novel experiment that investigates the utility of scanpaths in real-world applications, where imperfect viewing conditions are given.

## 1  Introduction

Vision enables humans and other animals the detection of conspecifics, food, predators, as well as other vital information even at considerable distances. However, a huge amount of information of about $10^7$ to $10^8$ bits reaches the optic nerve every second (Itti & Koch, 2001): to process it all would be impossible for biological hardware with limited computational capacity. The mechanism of visual attention allows to select only a subset of the visual information for further processing by means of eye movements. This breaks down the problem of scene understanding into a series of less demanding and localized visual analysis. The extracted subset of information mostly corresponds to the *fovea*, a circumscribed area of the retina highly populated with photoreceptors, which allows for fine vision (Bringmann et al., 2018). Fast eye movements called *saccades* serve to rapidly shift the fovea to regions of interest in the visual field (Purves et al., 2019). Between saccades, the direction of gaze remains still, during the so-called *fixation*, where visual information is collected. The sequence of fixations, also called *scanpath*, determines the flow of incoming information and is therefore critical for the effectiveness of human vision.

The selection performed by attention can be characterized by two distinct mechanisms: *bottom-up* and *top-down* (Connor et al., 2004). The bottom-up mechanism operates in the raw sensory input, rapidly

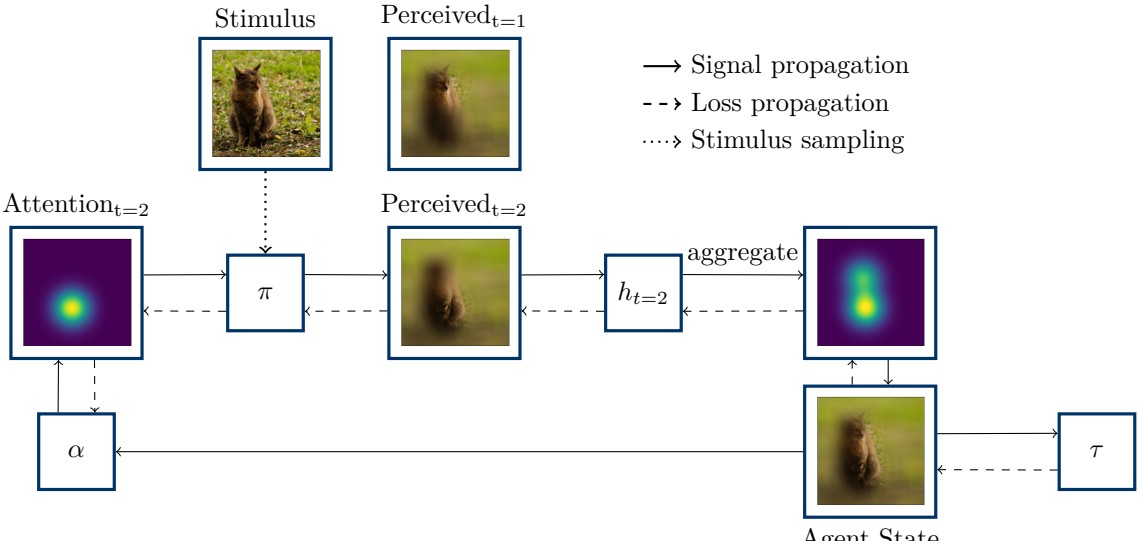

Figure 1: Illustration of the proposed Neural Visual Attention (NeVA) algorithm. A given *stimulus* is blurred and subsequently processed by the *attention mechanism* ($\alpha$). The attention mechanism calculates an *attention position* for the *foveation mechanism* ($\pi$). The foveation mechanism deblurs the image at the attention position and thus creates the *perceived stimulus* at time $t + 1$. The perceived stimulus is processed together with previously seen stimuli of the model to obtain the agent state of the image at time $t + 1$. Lastly, the loss signal of the *task model* ($\tau$) for the agent state is used to assess the quality of the attention position and thus guide and train the attention mechanism.

and involuntarily orienting attention towards visual features of potential importance (e.g., to a red spot in a green field). The top-down mechanism is slower, task-driven, and generally aimed at implementing long-term cognitive strategies (e.g., to assess the age of a person). Humans show a specific temporal pattern (Tatler et al., 2005): their very first fixations are generally guided by bottom-up features and exhibit high correlation, while they diverge very soon due to individual goals and motivations (i.e., tasks). This suggests that bottom-up attention can be described as a common mechanism among individuals, which makes it easier to be studied and characterized. For this reason, most recent studies have been oriented to bottom-up visual attention modeling.

Computational modeling of human visual attention lies at the intersection of many disciplines such as neuroscience, cognitive psychology, and computer vision. The construction of models that formalize the mechanisms of attention is of profound importance both for understanding the mechanism itself, potentially allowing the derivation of its biological implementation in the form of neural hardware (Koch & Ullman, 1987; Zanca et al., 2020a), and in the field of application such as surveillance (Yubing et al., 2011), video-compression (Hadizadeh & Bajić, 2013), virtual reality streaming (Rondon et al., 2021), image quality measure (Barkowsky et al., 2009), or question-answering (Das et al., 2017). Equipping artificial intelligence with biologically-inspired attention mechanisms could foster the robustness and interpretability of such systems (Luo et al., 2015; Vuyyuru et al., 2020). In the last three decades, many efforts have been made in developing computational models of visual attention. Based on the seminal work by Treisman describing the feature integration theory (Treisman & Gelade, 1980), current approaches assume a centralized role of the *saliency map*, i.e., a spatial map expressing the conspicuity of the information for each location in the visual field. Within this theory, attention shifts would be generated from the saliency map by the *winner-take-all* algorithm (Koch & Ullman, 1987), which selects the position associated with the highest saliency as the first fixation, then inhibits this location and moves to the second highest saliency location to generate the subsequent fixation, and so on. For this reason, the vast majority of studies have focused on improving saliency map estimation (Borji & Itti, 2012; Riche et al., 2013). Currently, the development of increasingly large eye-tracking datasets (Judd et al., 2009; Borji & Itti, 2015; Jiang et al., 2015), together with the use

of transfer learning techniques (Huang et al., 2015; Kümmerer et al., 2016; Borji, 2019), have made models based on deep convolutional neural networks the state-of-the-art in the saliency prediction task.

Following the trend of these works, the vast majority of eye-tracking datasets were collected under free-viewing conditions. This implies that subjects are required to "watch the images freely during the time they are presented" to them. This *should* eliminate the influence of any tasks and make the assumption of a bottom-up-driven vision (and attention) legitimate. However, this assumption drastically simplifies the problem of vision to that of a process of discovering *what is where* in the world. Instead, *vision* can not be thought of *just* as a information-processing task (Marr, 2010): our brains must be able to develop internal representations which serve as basis for future actions. This duality (information processing and representation learning) inextricably links vision to the concept of *intent* or, similarly, *task*. For example, even when looking at stimuli under free-viewing conditions, we still observe that objects drive attention better than bottom-up saliency (Einhäuser et al., 2008), or that emotional content influences attention dynamics already in the beginning of the visual exploration (Pilarczyk & Kuniecki, 2014). This leads to the conclusion that eye-tracking data collected under free-viewing conditions *may not* be free of task-driven components and the development of purely task-driven methods may help to quantify this phenomenon.

Most datasets and methods are designed for the purpose of saliency prediction (Boccignone et al., 2019b; Zanca et al., 2020b). However, saliency maps are static and do not encode the temporal aspect of visual exploration. Saliency maps do not take into account temporal dynamics, i.e., how fixations are ordered in a sequence. This implies that once collected, fixations are inherently interchangeable with respect to their temporal order. Consequently, the temporal order of fixations is largely ignored in experimental evaluations, with important consequences: 1) two scanpaths visiting the exact same points in a stimulus produce the same saliency map but may describe very different dynamics (e.g., some plausible, and some very unnatural). 2) Modelling these dynamics correctly can lead to more interpretable algorithms. To solve this issue, methods have been proposed that try to model the human scanpaths directly. Scanpath models can be separated into unsupervised, i.e., those that do not use any eye-tracking data to learn attentional shifts and supervised approaches. Since supervised methods (Jiang et al., 2015; Assens et al., 2018) cannot provide an explanation about the mechanism underlying human attention, they tend to overfit on the experimental setup of the available data, and hence generalise poorly across domains (i.e., require retraining) (Arjovsky et al., 2019). We focus on unsupervised methods in what follows. The circuitry of winner-take-all (Koch & Ullman, 1987), coupled with the mechanism of inhibition-of-return, can be combined with any unsupervised saliency estimation method, such as Itti's saliency map (Itti et al., 1998), to generate scanpaths in an unsupervised manner. Boccignone & Ferraro (2004) describe gaze shifts as a realization of a stochastic process with non-local transition probabilities defined in a saliency field, that represents a landscape upon which a constrained random walk is performed. Attention is then simulated as a search mechanism driven by a Langevin equation whose random term is generated by a Levy distribution. In (Zanca & Gori, 2017; Zanca et al., 2019a), attention is described as a dynamic process where eye movements are driven by three basic principles: boundedness of the retina, attractiveness of brightness gradients, and brightness invariance. Similarly, (Zanca et al., 2019b) proposes a mechanism of attention that emerges as the movement of a unitary mass in a gravitational field, where specific visual features (intensity, optical flow and face masks) are extracted for each location and compete to attract the focus of attention. A drawback of current approaches in computational modeling of visual attention is assuming *perfect information* (Bringmann et al., 2018). It is well known that human vision operates on *imperfect* (i.e., foveated) information and how *actions* (e.g., choosing the location of the next focus of attention) are taken on this basis. In contrast, all the proposed models tacitly assume that the visual input in its full resolution is available at each time. Wang et al. (2011), for example, design a mechanism of foveated vision but, to predict the new location of interest, they compare the foveated input with the original stimulus and select the point of greatest discrepancy. Gravitational models of attention (Zanca et al., 2019b) extract a gravitational field from feature maps that depends on the distance from the focus of attention in a similar way to the distribution of photoreceptors in the human retina, but the calculation of such fields is still done on the original stimulus at each time. In contrast, testing models that rely solely on imperfect vision could have a twofold advantage: highlighting how the implementation of biologically-inspired constraints can aid human-like scanpath simulation and developing methods that can actually work under resource-limited conditions.

While imperfect vision and task guidance have not been explored in the modeling of human visual attention, they have been successfully studied in other application areas. Larochelle & Hinton (2010) described a model based on a Boltzmann machine with third-order connections that can learn to accumulate information from different fixations, showing that it is possible to achieve performance at least on a par with models trained on whole images for certain classification tasks. In (Denil et al., 2012), a model constrained with foveated vision is trained to predict where to look for a task of image tracking. Different from previous proposals, the reward function is modeled as a Gaussian process to expand the action space from the discrete space of fixations to a continuous space. Mnih et al. (2014) propose a recurrent neural network model that is capable of extracting local information from an image or video by adaptively selecting a sequence of regions to be processed at high resolution. The resulting model is not differentiable, and it is trained to maximize performance on downstream tasks by a reinforcement learning procedure. In (Xu et al., 2015) an attention model is learned for image captioning, where qualitative evaluation shows that the model is able to attend to relevant objects while generating the corresponding words.

In summary, we observe two main limitations in current approaches for computational modeling of visual attention. 1) All mentioned approaches are based on raw sensory signals (bottom-up) and do not incorporate any task guidance. 2) All of the mentioned methods still assume perfect information, meaning that the original stimulus in its full resolution is needed to perform any computations.

In this paper, we propose the Neural Visual Attention (NeVA) algorithm, a purely task-driven mechanism of visual attention which operates with imperfect (i.e., foveated) information to generate scanpaths. NeVA utilizes differentiable deep neural networks (i.e., pre-trained for image classification or image reconstruction) which convey the concept of visual task in the framework. We define a differentiable layer that simulates the human-like foveation mechanism, which allows for end-to-end training of an attention mechanism guided exclusively by the error signal of the reference task. Our goal is not to develop a biologically plausible model of human attention but rather to observe whether imposing biologically-inspired constraints on neural networks will lead to more human-like image explorations. Moreover, we show that a top-down signal can be effectively incorporated to condition the generated scanpaths. An illustration of the NeVA method is given in Fig. 1. We emphasize that eye-tracking data is *not* used during training, but only for the evaluation of the proposed methods. Our contributions can be summarized as follows:

- We use common metrics to measure similarity between simulated and human scanpaths, and introduce and motivate the *String-Based Time-Delay Embeddings* (SBTDE), an alternative metric to account for stochastic components in human behavior.

- In an extensive evaluation with multiple well-established eye-tracking datasets, we show that scanpaths generated by NeVA exhibit the highest similarity to humans compared to state-of-the-art unsupervised methods, as measured by several metrics.

- The flexibility of the proposed approach allows us to quantify the contribution of different tasks (classification or reconstruction) to the generated scanpaths, and the importance of partial vision in generating plausible scanpaths (by comparing NeVA with its *optimization*-based version which assumes perfect vision).

- Lastly, we propose a novel experiment where the utility of scanpaths to solve a downstream visual task on unseen data under the constraint of imperfect information is assessed. In contrast to prior results that mainly focus on similarity to human scanpaths, this experiment aims to assess the usability of the simulated scanpaths for practical applications. Here, we show that NeVA-generated scanpaths are more effective at solving downstream visual tasks on unseen datasets, opening up possibilities for using NeVA models in practical applications where agents have to operate with constrained resources.

## 2 Methods

### 2.1 Notation

Here we introduce the notation necessary for the following sections. In our experimental setup, an agent (human or artificial) is presented with a static stimulus for a certain period of time $\mathbb{T}$. Let $S$ denote a visual stimulus,

$$S \in \mathbb{R}^d$$

where $d$ indicates the dimensionality of the visual field. In the case of a digitized stimulus, i.e., $S$ is organized as a two-dimensional grid of equally spaced pixels, $d$ equals the number of pixels and their respective color channels. Suppose the agent's perception is *foveated* and let $\xi$ denote its scanpath, which is defined as a sequence of fixations over a period of time $\mathbb{T}$,

$$\xi = \{\xi_t\}_{t \in \mathbb{T}}$$

such that $\xi_t \in \mathbb{R}^2$ represents the fixation location at time $t \in \mathbb{T}$. Finally, we denote by $\pi$ the *perceived stimulus*, which is obtained, at each time $t \in \mathbb{T}$, by means of an attention-dependent foveation mechanism that is applied to the original stimulus $S$, i.e.,

$$\pi \colon \mathbb{R}^d \times \mathbb{R}^2 \to \mathbb{R}^d$$
$$(S, \xi_t) \mapsto \pi(S, \xi_t)$$

The foveation mechanism should be designed to reflect the distribution of photoreceptors around the fovea. We provide a differentiable implementation of a foveation mechanism in the following sections. In biological systems, the agent *does not* have direct access to the original stimulus $S$, mainly due to limited resources. All of the agent's computations, including any attentional behavior, are based on the sole *imperfect* information carried by the perceived stimulus $\pi(S, \xi_t)$. This means that, if we denote by $\alpha$ the agent's attention mechanism responsible of generating scanpaths $\xi$, we can generally formalize it as

$$\alpha(\pi(S, \xi_t), h_{t-1}) \mapsto \xi_{t+1},$$

where $h_{t-1}$ denotes the agent's internal state at time $t - 1$. This assumption is often violated in most approaches of computational modeling of visual attention, where perfect vision is assumed, for example when predicting saliency (Borji & Itti, 2012) or scanpaths (Kümmerer & Bethge, 2021).

### 2.2 Neural Visual Attention (NeVA)

We propose an algorithm, which we call NeVA (Neural Visual Attention), to generate visual scanpaths in a purely task-driven approach. This is achieved by exploiting the loss signal of differentiable vision models (task models), such as convolutional artificial neural networks (CNNs), to guide visual attention. NeVA imposes the constraint of foveated vision on the perception of these task models. This is achieved by inserting a differentiable foveation mechanism before the first layer of the respective model. The foveation mechanism filters the original input and preserves only a small area of high acuity at the currently attended position. Thus, the performance of the task model degrades, which increases the loss. We exploit the loss signal of the task model to define a mechanism of selective attention that optimally solves the visual task at hand (i.e., which regions of the image are most important to solve the given task and should be attended). The algorithm consists of the following three main components which are described in what follows. Fig. 1 provides an illustration of the overall process.

#### 2.2.1 Task model

The task model $\tau$ is a differentiable model trained to solve a visual downstream task. Let $\mathbf{S} = \{S_1, ..., S_N\}$ be a collection of input stimuli. We specifically consider two different downstream tasks: a supervised *classification* and an unsupervised *reconstruction* task. In the case of supervised classification, a set $\mathbf{Y} = \{y_1, ..., y_N\}$ of semantic labels is available, where $y_i$ represents the main class of objects contained in $S_i$.

Labels are generally provided by a human supervisor and $\tau$ is trained to learn a mapping from $\mathbf{S}$ to $\mathbf{Y}$, such that

$$\tau(S_i) = y_i, \forall i \in \{1, ..., N\}.$$

In the case of reconstruction, $\tau$ is trained to compress the information of $S_i$ to a smaller dimensional space (*encoding*) and to reconstruct $S_i$ from this compressed representation (*decoding*). Therefore, the target domain matches the input domain, i.e., $\mathbf{Y} = \mathbf{S}$. This process can be formally described as $\tau \equiv \tau_{dec} \circ \tau_{enc}$, where $\circ$ denotes the function composition operator.

$$\tau_{dec} \circ \tau_{enc}(S) = S,$$

where $\tau_{enc}(S) \in \mathbb{R}^{\bar{d}}$, $\bar{d} < d$ to avoid trivial solutions and induce the learning of meaningful features. Hereby, we denote by $\mathcal{L}(\tau(\mathbf{S}), \mathbf{Y})$ the loss function evaluating how well the model $\tau$ solves the given task. This information will be used as a signal to guide the attention mechanism. In our experiments, we train one neural network for each task and use common loss functions. In this context, we use the categorical-cross-entropy loss to train the classification model and the mean-squared error loss to train the reconstruction model.

### 2.2.2 Foveation mechanism

To simulate human perception with any artificial system, we design a mechanism that is inspired by the biological constraint of foveated vision (i.e., a center of high visual acuity and a coarse resolution in the periphery). Given a stimulus $S$, the foveation mechanism computes the perceived stimulus $\pi(S, \xi_t)$ as a foveated rendering of $S$ centered at the current focus of attention $\xi_t$. To ensure the mechanism is differentiable and, therefore, can be integrated with any differentiable models (e.g., neural networks) for end-to-end learning, we define it as follows.

Let $\tilde{S}$ be a coarse version of $S$, obtained by applying a convolution to the original stimulus $S$ to suppress any high-frequency components, i.e.,

$$\tilde{S} = S * G_{\sigma_p},$$

where $G_{\sigma_p}$ is a Gaussian kernel with zero mean and standard deviation $\sigma_p$. From a biological viewpoint, $\tilde{S}$ can be regarded as the *gist* (Sampanes et al., 2008; Oliva & Torralba, 2006) that is perceived by peripheral vision in the first milliseconds of stimulus presentation. Furthermore, we simulate how the visual acuity decreases from the fovea center towards the periphery by means of a Gaussian blob $G_{\sigma_\xi}(t)$, with mean $\xi_t$ and standard deviation $\sigma_\xi$. The value of $\sigma_\xi$ is chosen to correspond with the average radius of the fovea region in humans' eyes, i.e., about 2 degrees of visual angle (Bringmann et al., 2018). Finally, the perceived stimulus can be obtained as a linear combination of the original stimulus and its coarse version,

$$\pi(S, \xi_t) = G_{\sigma_\xi}(t) \cdot S + (1 - G_{\sigma_\xi}(t)) \cdot \tilde{S},$$

which can also be interpreted as the operation of partially removing the smoothing effect $(\tilde{S} - S)$ from the original stimulus $S$, based on $G_{\sigma_\xi}$ as

$$\pi(S, \xi_t) = \tilde{S} - G_{\sigma_\xi}(t) \cdot (\tilde{S} - S).$$

Here, $\cdot$ denotes element-wise multiplication. It is worth noting that there is no leakage of information from the original stimulus to subsequent layers, as the $G$-matrices are fixed.

### 2.2.3 Task-driven attention mechanism

We consider an agent that sequentially explores the stimulus through a foveation mechanism. The agent's attention $\alpha$ is responsible of selecting the next location to attend depending on the current perceived stimulus $\pi(S, \xi_t)$. In addition, we assume the agent's decision to be based on the past perceived stimuli. To incorporate past information, we define the internal agent's state $h_t \sim h(S, \xi_t)$ to express the cumulative perceived information, or *memory* of the system. This state could be modeled, e.g., as a recurrent neural network. However, since we aim to analyze the effect of foveated vision on the visual exploration, we keep this

component as simple as possible. Thus, we simply aggregate past Gaussian blobs, which effectively increases the area of high visual acuity after each fixation:

$$h\left(S, \xi_t\right) = G_\Sigma(t) \cdot S + \left(J_d - G_\Sigma(t)\right) \cdot \tilde{S},$$

with

$$G_\Sigma(t) = \left[\sum_{i=0}^{\infty} \gamma^i G_{\sigma_\xi}(t-i)\right]^{0:1},$$

where $[...]^{0:1}$ is an element-wise clipping operator with minimum 0 and maximum 1, and $J_d$ is a $d \times d$ dimensional unit matrix (matrix only filled with ones). We notice that

$$h\left(S, \xi_t\right) = \pi\left(S, \xi_t\right) + h\left(S, \xi_{t-1}\right)$$

and the computation can be made more efficient by using the following iterative update rule

$$G_\Sigma(t) = \left[G_{\sigma_\xi}(t) + (1-\gamma)G_\Sigma(t-1)\right]^{0:1}.$$

The parameter $\gamma \in [0,1]$ can be regarded as a forgetting coefficient: the closer $\gamma$ is to 1, the faster past information is *dissipated*. Finally, we model an attention mechanism as a function of the cumulated perceived stimulus, such that

$$\alpha\left(h\left(S, \xi_t\right)\right) \mapsto \xi_{t+1}.$$

As we aim at defining a purely task-driven mechanism of attention, we determine the new fixations as the optimal ones for the resolution of the given task. We propose an approach that uses the loss signal of the task model $\tau$ as an auxiliary signal to guide the attention mechanism. At each time step $t$, we calculate

$$\mathcal{L}(\tau\left(h\left(S, \xi_t\right)\right), y)$$

and train a neural network (attention model) to minimize it by unblurring the relevant parts of the input using the foveation mechanism. Here, the input of the attention model is the internal agent state $h(\cdot)$ and the output is an attention position $\xi_{t+1} \in \mathcal{R}^2$. Subsequently, the attention position is used by the foveation mechanism to calculate the next perceived stimulus $\pi_{t+1}$. At the first step, no attention position is available and we present the fully blurred stimulus to the attention model to generate the first attention position $\xi_1$. The individual steps are shown in Fig. 1. Here, the attention model is presented with the internal agent state $h(\cdot)_{t=1}$ (here $h(\cdot)_{t=1} = \pi_{t=1}$ since the model has only performed a single fixation). Subsequently, the attention position $\xi_{t=2}$ and perceived stimulus $\pi t = 2$ are calculated. Next, the internal representation is updated by linearly combining the old internal representation with the newly perceived stimulus.

During training of the attention model, $h(\cdot)_t$ is forwarded to the task model. The task model calculates a loss value with respect to the downstream task. Lastly, the loss is backpropagated through the differentiable foveation mechanism to the attention model to update its weights. This solution is very efficient at inference time since the task model can be discarded and a scanpaths can be generated by iteratively predicting the next fixation using only the attention model. Additionally, this approach benefits from efficient implementations of neural networks on GPUs, which enable fast and parallelizable processing.

## 3 Experimental Setup

### 3.1 Datasets

To provide a broad evaluation of the proposed method against baselines and competitors, we selected a collection of three different and well-established eye-tracking datasets. They differ in input characteristics and resolution, eye-tracking device, and stimulus semantics. All three datasets were collected under the free-viewing condition in order to minimise the top-down (task-driven) influence of the subjects.

The **MIT1003** (Judd et al., 2009) dataset consists of 1003 natural indoor and outdoor scenes that include 779 landscape images and 228 portrait images with a horizontal and vertical resolution of up to 1024 pixels. The dataset contains fixations of 15 subjects during 3 seconds of free-viewing.

The **Toronto** (Bruce & Tsotsos, 2007) datasets consists of 120 images of outdoor and indoor scenes wit a resolution of $681 \times 511$ pixels. In contrast to the other two datasets, a large portion of the images in the Toronto datasets have no particular region of interest. The dataset contains fixations of 20 subjects during 4 seconds of free-viewing.

The **KOOTSTRA** (Kootstra et al., 2011) dataset contains 99 photographs from 5 categories with a resolution of $1024 \times 768$ pixels. This includes symmetrical natural objects, animals in a natural setting, street scenes, buildings, and natural environments. The category of natural environment is over-represented, with respect to the other categories. The dataset contains fixations of 31 subjects during 5 seconds of free-viewing.

### 3.2  Baselines and competitors

We restrict our choice of competitors to other unsupervised models, i.e., those models that do not learn an attention mechanism directly from human data. We point out here that in fact, in this work, eye-tracking datasets were only used for evaluation purposes and not during the learning of model parameters.

- Constrained Levy Exploration (CLE) (Boccignone & Ferraro, 2004). Scanpath is described as a realization of a stochastic process. We used saliency maps by (Itti et al., 1998) to define non-local transition probabilities. Langevin equation whose random term is generated by a Levy distribution, and a Metropolis algorithm, are used to generate scanpaths. We use the python implementation provided by the authors in (Boccignone et al., 2019a).

- Gravitational Eye Movements Laws (G-EYMOL) (Zanca et al., 2019b). Scanpath is obtained as the motion of a unitary mass within gravitational fields generated by salient visual features. We use the python implementation provided by the authors in the original paper.

- Winner-take-all (WTA) (Koch & Ullman, 1987). We use saliency maps by (Itti et al., 1998) and implement the algorithm as follows. We generate the first fixation at the location of maximum saliency. We apply inhibition-of-return by canceling the saliency in a neighborhood of the attended location in a ray of one degree of visual angle. The second fixation is selected as the location of maximum saliency in the updated map, and so on.

Additionally, some baselines were defined to better position the effectiveness of the approaches under investigation against, e.g., well-known human vision biases.

- Random baseline. Scanpaths are generated by subsequently sampling fixation points from a uniform probability distribution defined over the entire stimulus area. At each time step, each pixel of the digitized stimulus has the same probability of being attended to, regardless of its position or content.

- Center baseline. Scanpaths are randomly sampled from a 2-dimensional Gaussian distribution centered in the image. We used the center matrix provided in (Judd et al., 2009).

- Human baseline. This baseline is used to assess how well a human subject is predictive of another subject's attentional dynamics. Thus, for all metrics and datasets considered in this study, this baseline is calculated as the average obtained by considering each of the subjects relative to the rest of the population.

Lastly, we compare all approaches to the proposed NeVA (Neural Visual Attention) method. We test two different NeVA configurations. For the first configuration, we use a classification task model to supervise the NeVA-based attention model during training ($\text{NeVA}_\text{C}$). Here, we use a wide ResNet that was trained on the CIFAR10 dataset (Krizhevsky, 2009) from the RobustBench library. For the second configuration we use a reconstruction-task model ($\text{NeVA}_\text{R}$). In this case, we train a denoising autoencoder on the CIFAR10 dataset using the implementation proposed in (Zhang et al., 2017) and visually inspected the results to ensure that the training was conducted correctly. For both attention models we use wide ResNets (Zagoruyko & Komodakis, 2016). More details about the model configurations and training are given in Section A of the supplement.

### 3.3 Metrics

To obtain a quantitative measure of the similarity between two scanpaths, we use two different metrics. Let $A$ and $B$ denote two agents, observing the same stimulus $S$ during a certain period of time, and let $\xi^A = (\xi_1^A, ..., \xi_N^A)$ and $\xi^B = (\xi_1^B, ..., \xi_N^B)$ be their respective scanpaths. We only consider scanpaths of the same length $N$.

#### 3.3.1 String-edit distance (SED)

The string-edit distance (SED) (Jurafsky & Martin, 2000), also known as Levenshtein distance, allows to quantify the distance between two strings (i.e., sequences of letters) as the number of operations needed (deletion, insertion and substitution) to transform one string into the other one. It has been adapted in the domain of visual attention analysis to compare visual scanpaths (Brandt & Stark, 1997; Foulsham & Underwood, 2008). The stimulus $S$ is divided into $n \times n$ equally spaced regions, each of them uniquely labeled with a letter,

$$S = \bigcup_{i \in \{1, ..., n^2\}} S^{\vec{a_i}},$$

where $S^{\vec{a_i}} \subset S$, $S^{\vec{a_i}} \cup S^{\vec{a_j}} = \emptyset, \forall i \neq j$. Here, $a_i$ is the letter associated with the subset $S^{\vec{a_i}}$. Let *string* be the function that converts a scanpath to a string by assigning to each of the fixations the corresponding letter associated with the region the fixation belongs, i.e.,

$$string(\xi) \mapsto (a_{i_1}, ..., a_{i_N}),$$

where $\xi_j \in S^{\vec{a_{i_k}}}, \forall k \in \{1, ..., N\}$. Then, we can compute the string-edit distance between two scanpath as

$$SED\ \left(string(\xi^A), string(\xi^B)\right)$$

The string-edit distance has proven to be robust with respect to image resolution (provided the dictionary length is fixed for all stimuli) (Choi et al., 1995).

#### 3.3.2 String-based time-delay embeddings (SBTDE)

Previous research suggested as human responses to stimuli are not deterministic and people generally attend to different locations on the same stimulus. To take into account the stochastic component of the attention process (Pang et al., 2010), we would like to incorporate in our evaluations invariance with respect to delays. To this end, we define a metric that is based on the concept of time-delay embeddings (Abarbanel et al., 1994), widely use in physics to compare stochastic and dynamic trajectories. Time-delay embeddings have already been adopted for the analysis of visual attention scanpaths (Wang et al., 2011; Zanca et al., 2019b). By computing time delay embeddings, we can normalize the metrics with respect to the scanpath length. In order to make results robust with respect to changes in both stimulus resolution and scanpath length, we propose a modified version called string-based time-delay embeddings (SBTDE) that computes the time delay embedding in the string domain. Again, let $\xi^A$ and $\xi^B$ be two scanpath of the same length to be compared. Let $k \in \{1, ..., N\}$ be a reference sub-sequence length, we denote with

$$\Xi_k^A = \{(\xi_i, ..., \xi_{i+k})\}_{i \in \{1, ..., N-k\}},$$

the set of all possible sub-sequences of length $k$ contained in the scanpath $\xi^A$. Analogously, we define $\Xi_k^B$ for $\xi^B$. Then, we compute

$$d_k(x, \Xi_k^B) = \min_{y \in \Xi_k^B} \{SDE(string(x), string(y))\}.$$

In other words, given any sub-sequence $x$ of the scanpath $\xi^A$, we look for the sub-sequence $y$ of $\xi^B$ of minimal string-edit distance. Finally, we compute the $SBTDE_k$ of order $k$ as the average minimal distance, i.e.,

$$SBTDE_k(\xi^A, \xi^B) = \frac{1}{N} \sum_{x \in \Xi_k^A} d(x, \Xi_k^B).$$

### 3.3.3 Metrics visualization

To better understand the predictive power of the models and baselines under examination, we plot metrics at all possible sub-sequence length.

To take into account the large variability between human subjects, we calculate two types of averages for the metrics. For the first, which we will simply refer to as the score *mean*, for each stimulus, we compute the mean score of the scanpath under examination relatively to all human scanpaths available for that stimulus and then compute the mean for all stimuli in the dataset. In the second case, we make use of the recently proposed ScanPath Plausibility *SPP* (Fahimi & Bruce, 2021), which considers only the human scanpath with the minimum distance for each stimulus, and then computes the mean for all stimuli in the dataset. This metric demonstrates to be particularly useful in cases where a large number of different strategies is implemented by different subject (e.g., complex images with many candidate focal points).

Since longer scanpaths are more difficult to predict, the metrics might vary a lot with respect to the considered sub-sequence length. For this reason, we visualize the results with respect to the worst and best references, i.e., the *Random* and *Human* baselines.

### 3.4 NeVA forgetting hyperparameter

In a preliminary experiment, we analyzed the influence of the forgetting parameter $\gamma$ on the properties of the generated scanpaths on the MIT1003 dataset. For both NeVA configurations (classification and reconstruction supervision) setting $\gamma = 0.3$ let to the best results in all metrics. Thus, we set $\gamma = 0.3$ for the remaining experiments. A more detailed analysis is given in Section C of the supplement.

## 4 Results

In the following, we summarize the results of five different experiments used to evaluate the proposed NeVA method.

### 4.1 Simulating human visual attention under biologically-inspired constraints

The proposed NeVA attention model only considers the currently perceived stimulus $\pi\left(S, \xi_t\right)$ to generate the next attention position. To evaluate the effect of this property we propose an alternative approach, where we optimize the attention position $\xi_t$ directly to minimize the loss of the task model and consider multiple foveation positions simultaneously to generate the next attention position. In this case, we first calculate the loss of the task model for the currently perceived stimulus. Then, we compute the gradient with respect to the associated attention position $\xi_t$ and iteratively update the position using gradient descent. An efficient implementation of this optimization algorithm is described in Section D of the supplement.

To now asses the impact of using only imperfect information we compared the scanpaths generated by the scanpath model with those generated by the scanpath optimization algorithm on the MIT1003 dataset. We use the two different downstream-task models that were also used to train the NeVA attention models (NeVA$_\mathrm{C}$ and NeVA$_\mathrm{R}$) for the NeVA optimization algorithm described above. Additionally, we tested classification and reconstruction task models that were trained on ImageNet for the optimization-based NeVA algorithm (for all classification tasks we use pretrained models from the RobustBench library). We refer to these approaches as NeVA-O$_\mathrm{C}$ and NeVA-O$_\mathrm{R}$. Detailed information about the downstream-task models are given in Appendix D. In our experiments on MIT1003, the three classification-based NeVA approaches achieve the lowest SBTDE distance for all scanpath lengths. Nevertheless, the trained attention model demonstrates better performance than the two optimization-based approaches. The two CIFAR10 reconstruction-based approaches show similar metrics, while the ImageNet reconstruction-based approach is considerably worse than all other approaches. We refrained from training ImageNet-based NeVA attention models for the remaining experiments due to the following two reasons. Scanpaths that were generated with ImageNet-based supervision showed a higher mean and SPP-SBTDE distance in our experiments than those created with CIFAR10-based supervision. Additionally, CIFAR10-based attention models are substantially faster at the inference time, since we can re-scale images to the relatively low training resolution of $32x32$ before

processing. The scanpath metrics of the different configurations are summarized in Figure 8 in Appendix 4.1. We discuss possible explanations for the differences between CIFAR10 and ImageNet-based models and the disadvantages of the optimization-based approach in Section 5.

## 4.2 Scanpath plausibility

We evaluated the similarity of scanpaths generated by NeVA to those of human subjects on three different datasets. For each stimulus, NeVA is used to generate a task-driven visual exploration scanpath. The generated scanpaths were compared with human scanpaths collected by eye-tracking. The performance of NeVA was compared with those of three state-of-the-art unsupervised scanpath models (Boccignone & Ferraro, 2004; Zanca et al., 2019b; Koch & Ullman, 1987) of visual attention and three baselines (random, center, and human baselines). We generate scanpaths of length 10 (i.e., sequences of 10 fixations) with all models and baselines. We only consider the first 10 fixations of human scanpaths and discard recordings with less than 10 fixations. Nevertheless, we investigate the behavior of early fixations by analyzing the similarity of scanpaths over time. Table 1 summarizes the results for all metrics and datasets. Scanpaths generated by the $NeVA_C$ are best on average over all metrics for all datasets. For individual metrics $NeVA_C$ is the best-performing method in 11/12 metrics and the second-best in the remaining 1. $NeVA_R$ is the second best method in 7/12 metrics. G-Eymol is the second-best method in 4 metrics, while CLE is the best method in 1 metric and the second-best method in 1 metric (same score as G-Eymol). Figure 2 shows the similarity of artificially generated and human scanpaths for the three different datasets.

For the **MIT1003** dataset, scanpaths generated by the $NeVA_C$ approach demonstrate the overall highest similarity to human scanpaths. Specifically for long scanpaths, the mean distance and SPP distance (distance to the scanpath of the closest human subject) are considerably lower than those of all other competitors. $NeVA_R$ is the second-best approach in terms of the SBTDE distance and is comparable with G-Eymol in the SED. The bottom-up models (G-Eymol and CLE) perform particularly well in the very first fixations. The scanpaths generated by all competitor approaches exhibit considerably higher similarity to human scanpaths than the two simple baselines (random, center) and the basic approach of WTA.

For the **Toronto** dataset, the results for the SED are similar to MIT1003. However, for this dataset CLE and G-Eymol show similar performance. The SBTDE distance is substantially lower for both NeVA approaches for all scanpath lengths compared to all competitors. Furthermore, for subsequence lengths of $k < 9$ the mean SBTDE distance of the NeVA approaches is smaller than the SBTDE distance between the human subjects. This might be due to the fact that being composed of generic outdoor and indoor images, the scanpaths in the Toronto dataset exhibit high variability. Hence a single subject is generally a bad predictor of the remaining population, especially in the short term.

For the **Kootstra** dataset, $NeVA_C$ does not achieve the lowest SPP-SED for a small number of fixations ($< 8$) but shows the best performance for long scanpaths compared to the other methods. Both NeVA approaches achieve a considerably lower mean SBTDE distance than all other approaches. In contrast to the other results, CLE demonstrates the lowest SPP-match SBTDE distance followed by the two NeVA-based approaches with a considerable gap to the other methods.

Example scanpaths generated with NeVA are shown in Fig. 3. A qualitative comparison with existing approaches is given in Appendix H.

## 4.3 Loss behavior during visual explorations

To determine the cause of the difference between classification and reconstruction-based supervised models, we further analyzed the loss behavior of the downstream-task models during scanpath generation. Note that we do not investigate the training loss, but rather how the loss changes for an already trained model during scanpath generation at inference time. Figure 4 demonstrates that the loss of the classification-task model rapidly decreases after the first fixation and remains largely the same for the remaining fixations. In contrast, the loss of the reconstruction task model decreases more consistently. We argue that the considerable decrease in the loss of classification task models after the first fixation leads to less exploration and shorter scanpaths. Equivalently, the slower decay of the loss in the reconstruction model leads to more exploration and longer

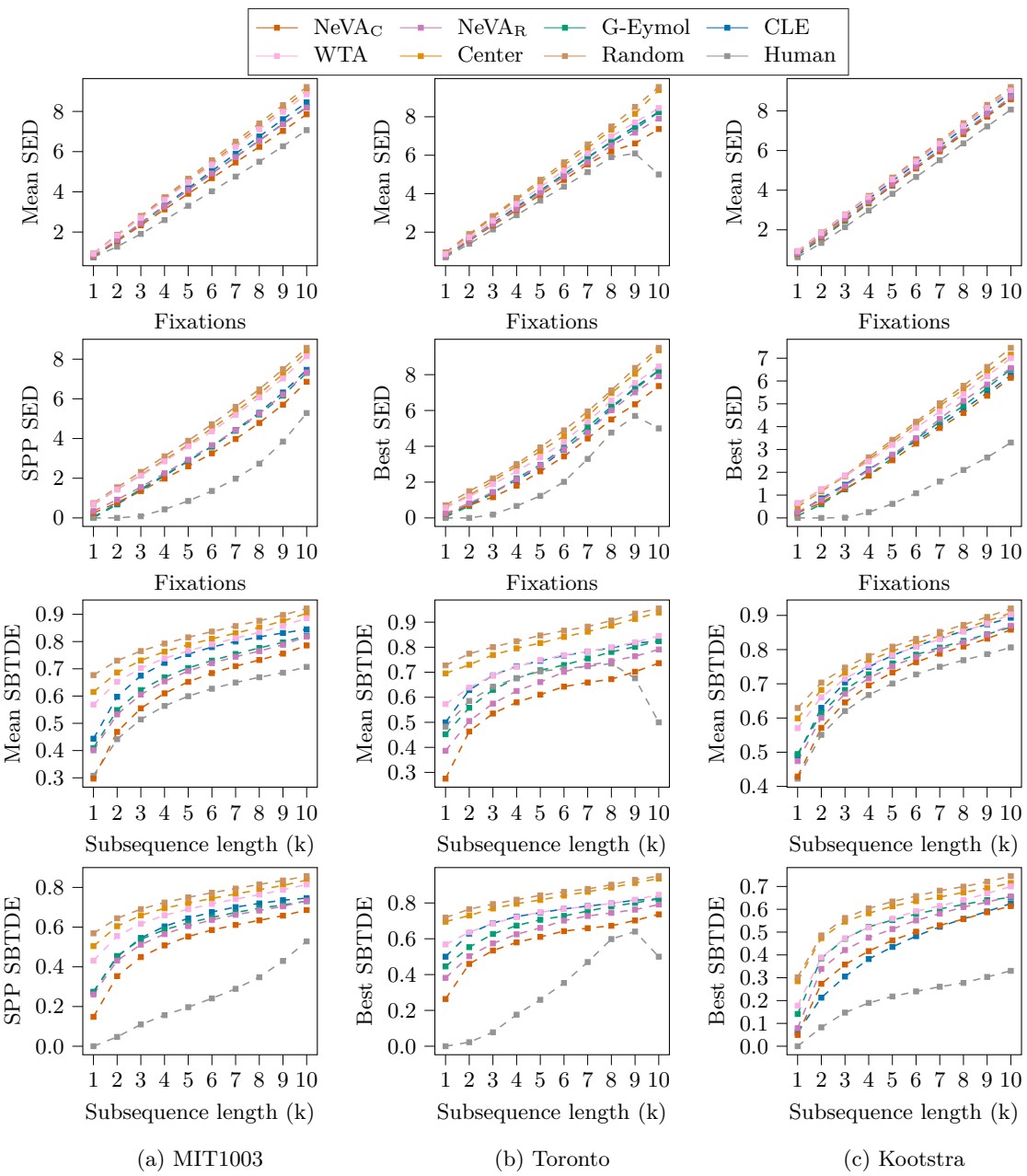

Figure 2: Similarity of human and artificially generated scanpaths from different methods assessed with 4 different distance metrics for the MIT1003 dataset. A **lower** score corresponds to a higher similarity to human scanpaths for each metric. The human baseline is calculated by comparing scanpaths between different subjects and can be considered as a gold standard.

Table 1: Similarity of scanpaths generated by the proposed NeVA method and other competitors to those of humans for several metrics. A lower score in each metric corresponds to a higher similarity to human scanpaths. The best results are shown in **bold** and the second best results are underlined. The human column denotes the intra-scanpath distance between humans.

| Datasets | NeVA$_C$ | NeVA$_R$ | G-Eymol | CLE | WTA | Center | Random | Human |
|---|---|---|---|---|---|---|---|---|
| **MIT1003** | | | | | | | | |
| Mean SED | **4.3** | 4.49 | 4.48 | 4.60 | 4.90 | 4.99 | 5.09 | 3.74 |
| SPP SED | **3.15** | 3.49 | 3.39 | 3.41 | 4.15 | 4.26 | 4.44 | 1.65 |
| Mean SBTDE | **0.62** | 0.67 | 0.68 | 0.72 | 0.76 | 0.78 | 0.81 | 0.57 |
| SPP SBTDE | **0.51** | 0.57 | 0.59 | 0.60 | 0.67 | 0.71 | 0.74 | 0.23 |
| **Average** | **2.14** | 2.30 | 2.28 | 2.33 | 2.62 | 2.68 | 2.77 | 1.55 |
| **Toronto** | | | | | | | | |
| Mean SED | **4.22** | 4.42 | 4.52 | 4.56 | 4.74 | 5.05 | 5.19 | 3.72 |
| SPP SED | **3.35** | 3.71 | 3.79 | 3.74 | 4.17 | 4.51 | 4.71 | 2.28 |
| Mean SBTDE | **0.58** | 0.64 | 0.69 | 0.72 | 0.73 | 0.82 | 0.85 | 0.64 |
| SPP SBTDE | **0.58** | 0.64 | 0.68 | 0.72 | 0.73 | 0.82 | 0.84 | 0.30 |
| **Average** | **2.18** | 2.35 | 2.42 | 2.44 | 2.59 | 2.80 | 2.90 | 1.74 |
| **Kootstra** | | | | | | | | |
| Mean SED | **4.66** | 4.75 | 4.67 | 4.89 | 4.99 | 4.99 | 5.08 | 4.26 |
| SPP SED | **2.98** | 3.26 | 3.12 | 3.12 | 3.66 | 3.75 | 3.88 | 1.16 |
| Mean SBTDE | **0.71** | 0.73 | 0.74 | 0.76 | 0.77 | 0.78 | 0.80 | 0.68 |
| SPP SBTDE | 0.43 | 0.48 | 0.51 | **0.41** | 0.53 | 0.58 | 0.60 | 0.20 |
| **Average** | **2.19** | 2.31 | 2.21 | 2.30 | 2.49 | 2.53 | 2.59 | 1.57 |

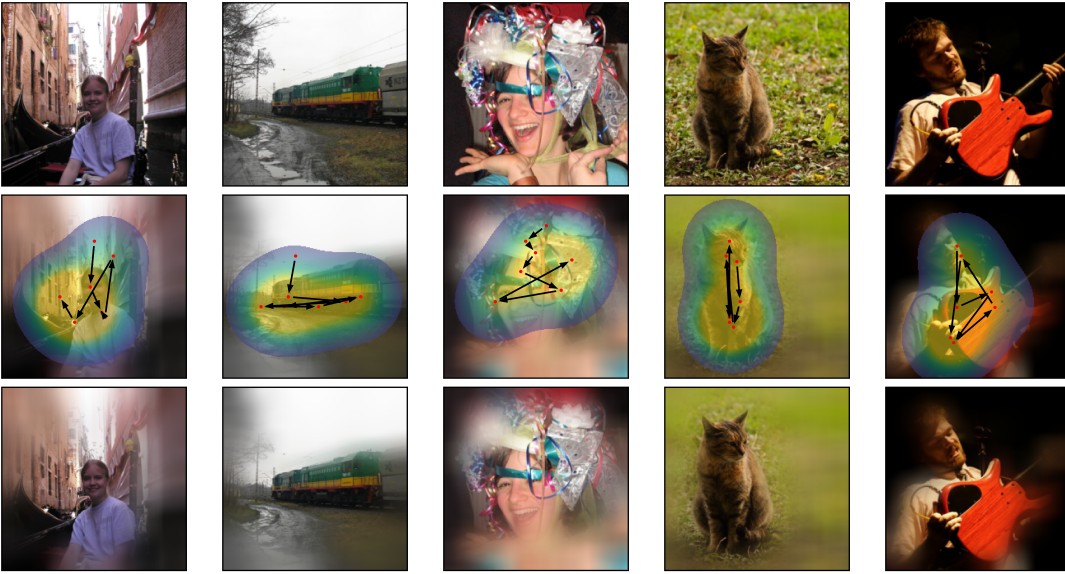

Figure 3: Example scanpaths generated by NeVA. The first row shows the original stimulus, the second row shows the foveation heatmaps and fixations, and the last row shows the internal representation after the last fixation.

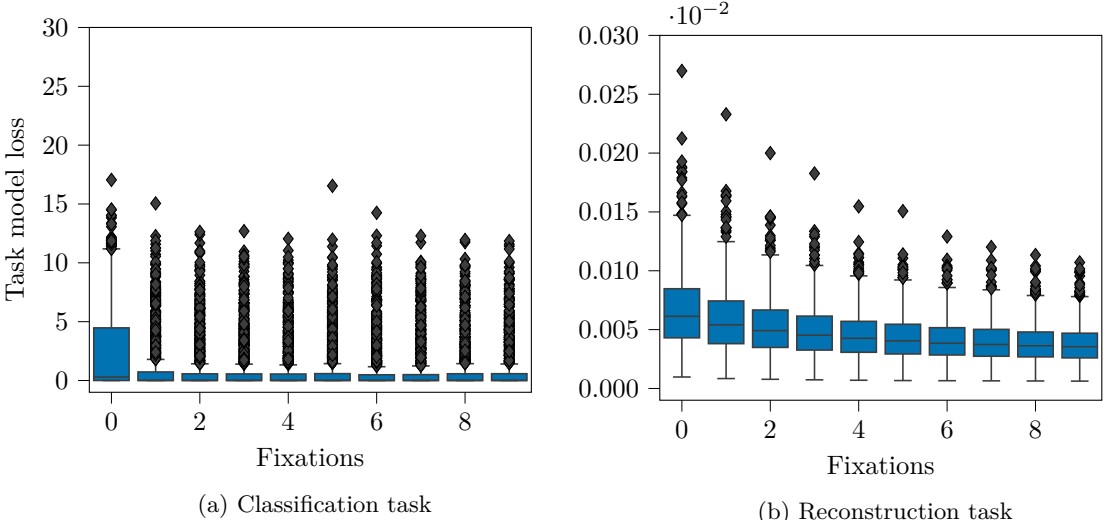

(a) Classification task

(b) Reconstruction task

Figure 4: Box plots of the task model loss during different fixations averaged over all datasets. The loss of a classification (a) and a reconstruction (b) task model is shown. Note that the difference in loss magnitude between the two tasks is caused by the two different loss functions (categorical cross entropy for classification and mean square error for reconstruction).

scanpaths. It is important to note that we do not expect a decrease in loss to necessarily correspond to more human-like scanpaths. Otherwise, the optimization-based approach would generate more human-like scanpaths.

### 4.4 Application in downstream tasks

In real-world scenarios, scanpaths can be used to reduce computational overhead of downstream applications (e.g., by reducing the resolution of foveated image regions). To assess the usability of the different scanpath methods for possible application scenarios we explored their ability to generate scanpaths for a classification downstream task on an unknown dataset with unseen classes. We evaluate the performance of the different methods on subsets of the CIFAR100 and ImageNet datasets (1000 images each). To measure the accuracy, we use two different ResNet classifiers trained on the complete CIFAR100 and ImageNet datasets, respectively. Both datasets were not seen by the downstream-task models used in the NeVA method that were trained on CIFAR10. The results are summarized in Table 2. The overall best performance is achieved by the two NeVA-based algorithms followed by the G-Eymol method for both datasets. These methods also showed the best performance for creating human-like scanpaths in Section 4.2. Note that NeVA achieves higher accuracy than all prior approaches, although it has only access to the current perceived stimulus while creating the scanpaths, while all other methods use the real stimulus. This property makes NeVA more suitable in real-world applications, where only an imperfect version of the stimulus might be available to save computational resources. The considerable accuracy degradation for the CIFAR100 dataset from clean images to partly blurred images indicates that the blurring of the images has a considerable effect even for low-resolution images. We observed that the overall accuracy of the classifier monotonically increased with the number of fixations (see Appendix E). However, since the CIFAR100 and ImageNet classifier were not trained to classify partially blurred images the accuracy converges only slowly to the value observed on clean images.

### 4.5 Influence of the predicted label

In the previous section, we observed quantitative differences between scanpaths generated with classification and reconstruction supervision. To further investigate the influence of the task model on the scanpaths, we explored the impact of the predicted label of a classification task model on the quality of the generated

Table 2: Accuracy of a classification task, where the input is a blurred version of the original input. The input is sequentially deblurred using scanpaths from different methods and the accuracy is averaged over the scanpath. Both subsets of the CIFAR100 and the ImageNet dataset (1000 images each) are considered. Best results are shown in **bold** and second-best results are underlined. For the proposed NeVA method C indicates an attention model trained on a classification downstream task and R an attention model trained on a reconstruction downstream task.

| Datasets | Standard | NeVA$_C$ | NeVA$_R$ | G-Eymol | CLE | WTA | Center | Random |
|----------|----------|----------|----------|---------|-------|-------|--------|--------|
| CIFAR100 | 57.40 | **39.68** | 38.23 | 37.95 | 30.99 | 25.92 | 33.41 | 25.66 |
| ImageNet | 75.50 | **61.03** | 58.89 | 55.95 | 52.96 | 47.65 | 55.72 | 54.07 |

scanpaths. It is not immediately clear how the predicted label influences the scanpath as the task model that was used for most experiments was trained on the CIFAR10 dataset, which does not contain most of the objects present in the different evaluation datasets (MIT1003, Toronto, and Kootstra). First, we assessed how less or more fine-grained labels of a classification task model affect the scanpaths generated by the NeVA method. Therefore, we generated scanpaths with two different NeVA configurations for two different images of the MIT1003 dataset. For the first configuration, we trained a NeVA attention model using a CIFAR10 classifier as task model. For the second configuration, we used the NeVA optimization algorithm to generate scanpaths directly with an ImageNet classifier. Figure 5 illustrates that more fine-grained labels can lead to less human-like scanpaths. The ImageNet classifier predicts very specific labels for the objects in the scene. This leads to a focus of the attention on these models (i.e., *"baseball player"* in (b) and *"dumbbell"* in (d)).

Next, we assessed how changing the label predicted by the classifier affects the generated scanpaths of the NeVA optimization algorithm. Szegedy et al. (2014) demonstrated that neural networks exhibit the intriguing property that their predictions can be easily influenced by so-called adversarial perturbations that are not visible to human perception. We utilized adversarial perturbations to change the predicted label of the task model by applying them to the original stimulus. We then compared the scanpaths generated under adversarial attacks to normally generated scanpaths. Additionally, we investigate if the resulting scanpaths achieve worse performance in the classification downstream task proposed in Section 4.4. Note that we generate the scanpaths using the NeVA optimization algorithm and not a NeVA-based attention model. This was done as the attention model is not dependent on the labels predicted by the task model during inference.

The adversarial perturbation is generated as follows. We aim to find a perturbation $\gamma$ that when added to the pixel values of the original stimulus $S$, lets the task model predict a different label for the adversarial stimulus $S_{adv} = S + \gamma$. Here, we constrain the $\ell_\infty$ norm of *gamma* to $||\gamma||_\infty < \varepsilon$, so that it does not influence the human perception. We set $\varepsilon = 4/255$ which was sufficient to reduce the accuracy of the classifier to below random guessing. To solve this optimization problem, we use the Jitter attack with 100 iterations (Schwinn et al., 2021b).

Scanpaths that are generated for adversarial attacked stimuli demonstrate less similarity to human scanpaths in both analyzed metrics. The analysis of the scanpaths in the presence of adversarial attacks is shown in Figure 10 of Appendix F. Moreover, the label change has a negative influence on the performance of generated scanpaths in an unknown classification downstream task. The accuracy is considerably reduced for both the subset of CIFAR100 (4.86%) and the subset of ImageNet (5.19%). After the attack, the simple center baseline achieves higher performance than NeVA on both datasets, although it does not consider any image information. The results indicate that the label has a considerable influence on the scanpath even if the originally predicted label does not make sense (this is the case for most of the MIT1003 images since the image labels do not overlap with the labels of CIFAR10, which the NeVA models were trained on). This suggests that the model encodes label-specific concepts in its predictions which affect the exploration in a substantial manner even for unknown objects. We further discuss this observation in Section 5. An overview is given in Table 4 of Appendix F.

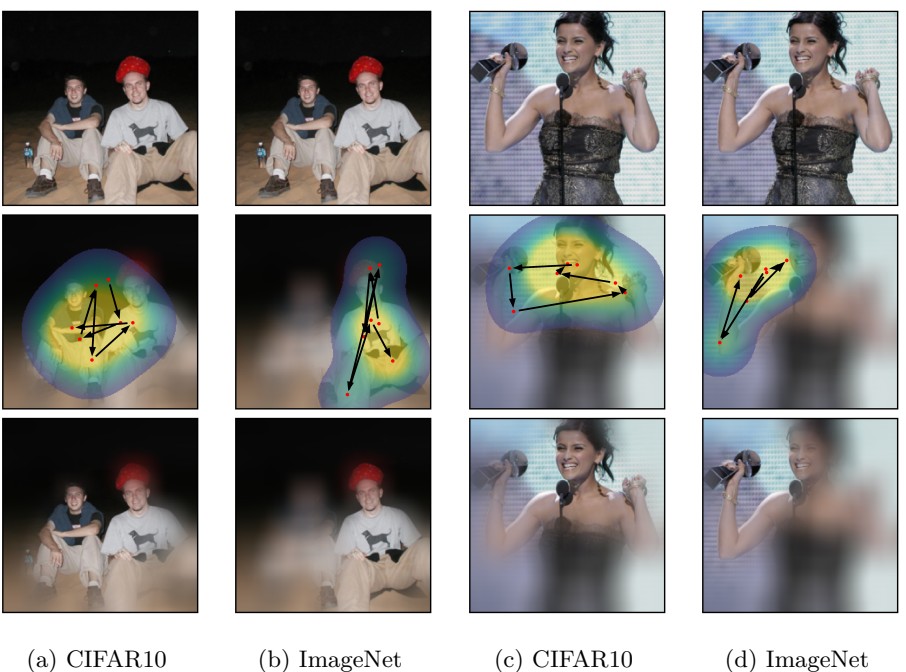

(a) CIFAR10    (b) ImageNet    (c) CIFAR10    (d) ImageNet

Figure 5: Scanpaths generated from the MIT1003 datasets with two different NeVA configurations. For the first NeVA configurations an attention model is trained under the supervision of a CIFAR10 classification model. For the second configuration scanpaths are directly generated using the NeVA optimization algorithm and an ImageNet classifier. The greedy optimization of scanpaths with the ImageNet model leads to locally constrained scanpaths. This is caused by the labels predicted by the ImageNet classifier. For the scanpath generated in (b) the model predicts *"baseball player"* and focuses only on the person with the hat. For the scanpath shown in (d) the model predicts *"dumbbell"* and focuses mainly on the arm and object in the hand of the woman.

### 4.6 Additional analysis

To gain additional quantitative insights into the scanpaths generated with NeVA, we conducted further analyses. In section G we observe that the NeVA algorithm exhibits greater similarity to humans in terms of the distribution of saccade amplitudes, followed by G-Eymol. Note that none of the models directly enforce a preference for the distance between successive fixations. NeVA models supervised by the reconstruction tasks tend to have wider explorations, as they manage to decrease the task model loss progressively with each fixation. A different observation is made for classification task-driven models, which minimizes the loss already at the first fixation (see Figure 4). These loss results are shown in sections G and 4.3.

## 5    Discussion

In this section, we discuss results from the previous section and relate them to existing work from the literature.

**Top-down signal guidance explains human attention better than any existing bottom-up approach.** We conducted extensive experiments to compare NeVA with competitors (WTA, CLE, and G-Eymol) and baselines (Random, Center, and Humans). Unlike competing approaches that are mainly bottom-up, NeVA is driven solely by the signal obtained from the task model and scanpaths are generated in order to maximize performance in a top-down manner. We found that top-down guidance explains human scanpaths in terms of scanpath similarity considerably better than bottom-up approaches. However, in line with previous research demonstrating bottom-up guidance in the very first phase of visual exploration, in the first few fixations, NeVA is often worse than existing bottom-up approaches. This observation motivates the combination of top-down guidance by neural networks and bottom-up mechanisms in future work. The Center baseline did not perform much better than Random. This is a surprising result if we consider that a center blob can predict saliency very well (Judd et al., 2009; Borji & Itti, 2012). Additionally, the result reinforces the notion, that compared to saliency prediction, the modeling of a scanpath mechanism is a more complex problem. On the other hand, we could observe that Human baseline (obtained by comparing one subject each time with the rest of the population) largely outperforms each model, especially in the SPP-metrics. These observations have two important implications. First, there is still room for improvement in the computational modelling of visual exploration dynamics. Second, to some extent, it is possible to identify within the human set clustered diversity: the fact that the SPP-metric is significantly better than the mean metric suggests that, for every human, there is at least one human which explores the scene in a comparable way. A similar result was obtained by (Zangrossi et al., 2021). This clustered diversity needs to be further analyzed to lead to better and more diverse models. One solution could be the introduction of non-deterministic components into the models. Another approach could consist in modeling the mechanisms that lead to the diversity within humans with hyperparameters (e.g., exploration tendency, or underlying task) to simulate different persons and behaviors. The classification task consistently led to better performance in terms of scanpath similarity compared to the reconstruction task. This could strengthen the hypothesis that, even in free-viewing conditions, humans tend to be guided by objects (Einhäuser et al., 2008) before considering any other information in the scene.

**Imperfect information benefits the generated visual behaviors.** In this work, we took special care on modeling imperfect vision, i.e., to ensure that the attention model never has direct access to the original stimulus in its full resolution (but only through foveation). This is because, in addition to not having biological validity, the assumption of perfect vision could actually disfavor modeling by placing the artificial system in conditions different from the human ones it aims to imitate. The results presented in Figure 8 suggest that imposing biologically-inspired constraints, in fact, benefits the resulting generated behaviors. The optimized version of the model, i.e., NeVA-O, selects the next fixation after having actually observed all possible locations in the visual field. This resulted in a less plausible behavior when compared with humans. We claim that the optimized version of the model will avoid purely exploratory fixations which are needed by a system with no perfect information to minimize its uncertainty toward the environment.

**Top-down features versus task-driven approaches.** The contribution of top-down features to attention modeling has been extensively investigated in the literature (Kummerer et al., 2017; Borji, 2012). Independently, models have been proposed that instead implicitly learn an attention mechanism in order to solve a given task (Dosovitskiy et al., 2020; Xu et al., 2015) (with no relation to human attention modeling). To the best of our knowledge, however, NeVA is the first attempt in which the latter is designed to incorporate biologically-inspired constraints, simulate human attention, and evaluated the mechanism in terms of similarity to human scanpaths.

**Performance on resource-constrained real-world applications.** The goal of computational modeling of visual attention is not only to improve our understanding of human mechanisms but to develop algorithms that can equip any measurement-constrained autonomous exploration system with similar capabilities, e.g., a space rover which needs to perform region-of-interest prioritize sampling (Bhattacharjee et al., 2021).

In order to use an artificial attention mechanism in real-world applications, it has to be computationally efficient, i.e., by being subject to the biologically-inspired constraint of foveated vision or usable with low-resolution data. Otherwise, the computational cost of generating the scanpath may negate any potential savings in processing resources. In contrast to existing work (Borji & Itti, 2012; Kümmerer & Bethge, 2021), NeVA generates fixations by only considering the currently perceived foveated stimulus. To evaluate the usability of scanpaths in real-world tasks we propose a novel experiment where the utility of scanpaths to solve a downstream classification task is analyzed. This form of evaluation differs notably from previous work in the area of human visual attention, in which scanpaths were evaluated primarily on their similarity to human scanpaths. However, similar approaches can be found in the domain of photon-counting sensors where systems need to make decisions based on a partial version of the input (Chen & Perona, 2016). Our experiments showed that, although NeVA does not use the original stimulus for scanpath generation, it still outperforms prior methods in the downstream task of object classification with unknown object categories. Moreover, since the used NeVA model was trained with CIFAR10-based supervision it takes low-resolution images in $\mathbb{R}^{32 \times 32}$ as an input instead of the full-resolution images. This further considerably reduces the computational requirements. In future work, joint training of both the task model and the NeVA attention model in an end-to-end fashion might further increase the utility of the generated scanpaths. This would prompt the task model to focus on important features in the image since it would have to make predictions based solely on the currently perceived stimulus. In turn, attending the correct positions would become even more crucial for solving the task, which might improve the attention model.

**On the relationship between deep neural networks, gradients, and human attention.** Additional motivation for the proposed NeVA algorithm can be found in the research area of explainable machine learning. Prior work showed that the gradients of machine learning models with respect to the input can be used to identify important features in the data (Baehrens et al., 2010; Simonyan et al., 2014; Sundararajan et al., 2017). Furthermore, gradient-based attribution methods have shown to be aligned with human semantics (Selvaraju et al., 2017; Smilkov et al., 2017). Besides, another line of research showed that gradients of neural networks can be used to identify untrustworthy data (Schwinn et al., 2021a) (e.g., semantically meaningless data such as noise or out-of-distribution data). These findings further motivate the use of gradients from a task model to guide visual attention in our approach.

**Trade-off between correct and too specific (downstream) classification.** First and foremost, current literature suggests that a purely top-down description of visual attention is not biologically plausible (Treisman & Gelade, 1980; Duan & Wang, 2015). This is not, in fact, a goal of this paper, which is only aimed at demonstrating how, within the proposed evaluation scheme, top-down components have a prominent role in explaining the vast majority of attentional shifts. As we already suggested, an integration of bottom-up components in the current method is worth investigating and could bring a more comprehensive explanation of the phenomenon of attention. In fact, our experiments already highlight some limitations of top-down attention. Namely, as the proposed attention mechanism is highly dependent on the task model that guides it, it also inherits its limitations. In our experiments, the supervision of an ImageNet-based classifier (i.e., trained to recognize objects among 1000 classes) lead to highly class-specific scanpaths that explored only highly specific regions of the respective images. An ImageNet classifier must learn to distin-

guish 1000 different classes that may have semantic similarities (e.g., multiple dog breeds). We argue that the classifier thus learns to focus on descriptive local features and the scanpath becomes highly specific. In contrast, CIFAR10-based classifiers learn more high-level features as the classes in CIFAR10 are semantically more different. This problem might be circumvented by considering task models that can learn more comprehensive representations of the world. This includes object detectors like YOLO (Redmon et al., 2016) that can detect multiple objects at the same time and thereby should not saturate as fast as classifier-based task models (see Figure 4). Other options could be multi-label classification tasks (Dembczynski et al., 2010), models that perform hierarchical classification (Marszalek & Schmid, 2007) and can attend broader and fine-grained concepts at the same time, or a weighted combination of different tasks. Lastly, we demonstrated that attention models can generalize well even if the supervising task model does not contain the concepts present in the image. This hypothesis was further strengthened by the fact, that changing the predicted label of the task model had a considerable influence on the quality of the generated scanpaths in terms of similarity to human scanpaths and applicability in downstream tasks.

**Study implications and limitations.** The key takeaway of this work is that adding biologically-inspired constraints to neural networks makes them explore images in a more human-like manner. We believe that this finding can be integrated into existing approaches that model human attention. Moreover, it motivates further studies on other biological constraints and if adding them to neural networks further improves the alignment between artificial and human scanpaths. In this first work, we focus on foveated vision as it is present at the very first steps of the vision "pipeline". We believe that having limited access to visual input is crucial to develop a plausible attention mechanism. For the sake of a clear interpretation of the results, we have limited ourselves to this component in the first experiments. However, it is of great interest to extend it to other biological constraints (e.g., eye-fixation duration) in future work.

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

# A Model Training

We trained multiple different neural networks in the scope of this work. The training hyperparameters and choices are given in the following sections.

## A.1 Denoising autoencoder

We trained two denoising autoencoders (Zhang et al., 2017) using the implementation proposed in (Zhang et al., 2017) on the CIFAR10 (Krizhevsky, 2009) and ImageNet datasets (Deng et al., 2009). These models were used to supervise the training of attention models as described in Section 2.2.3 or for the optimization algorithm described in Section 4.1. We used 17 layers and set the number of output channels of the first layer to 64. We used an Adam optimizer with a learning rate of $l = 0.001$ for both datasets. This was found to be the best value for CIFAR10, as found by a small grid search for $l \in \{0.0001, 0.001, 0.01\}$. We trained the networks for 100 and 5 epochs for CIFAR10 and ImageNet, respectively. This was sufficient for the convergence of the loss in both cases. For both datasets, we used a batch size of 100. For CIFAR10 we injected noise from a Gaussian distribution with a standard deviation of 0.15 and for ImageNet we used a standard deviation of 0.3.

Table 3: Similarity of scanpaths generated by the proposed NeVA method and other competitors to those of humans for several metrics. Results are **normalized** such that the average score between humans equals **1** and the average score between humans and random scanpaths equals **0**. A lower score in each metric corresponds to a higher similarity to human scanpaths. The best results are shown in **bold** and the second best results are underlined.

| Datasets | NeVA$_C$ | NeVA$_R$ | G-Eymol | CLE | WTA | Center |
|---|---|---|---|---|---|---|
| **MIT1003** | | | | | | |
| Mean N-SED | **0.41** | 0.55 | 0.50 | 0.58 | 0.88 | 0.92 |
| SPP N-SED | **0.51** | 0.63 | 0.57 | 0.57 | 0.90 | 0.93 |
| Mean N-SBTDE | **0.22** | 0.41 | 0.45 | 0.64 | 0.77 | 0.87 |
| SPP N-SBTDE | **0.55** | 0.67 | 0.70 | 0.73 | 0.86 | 0.93 |
| **Average** | **0.42** | 0.57 | 0.55 | 0.63 | 0.85 | 0.91 |
| **Toronto** | | | | | | |
| Mean N-SED | **0.30** | 0.42 | 0.44 | 0.49 | 0.67 | 0.89 |
| SPP N-SED | **0.42** | 0.56 | 0.56 | 0.55 | 0.78 | 0.90 |
| Mean N-SBTDE | **-0.4** | 0.0 | 0.14 | 0.35 | 0.39 | 0.85 |
| SPP N-SBTDE | **0.48** | 0.61 | 0.69 | 0.76 | 0.77 | 0.95 |
| **Average** | **0.20** | 0.37 | 0.46 | 0.53 | 0.65 | 0.90 |
| **Kootstra** | | | | | | |
| Mean N-SED | 0.50 | 0.60 | **0.47** | 0.78 | 0.89 | 0.88 |
| SPP N-SED | **0.63** | 0.73 | 0.65 | 0.69 | 0.94 | 0.93 |
| Mean N-SBTDE | **0.28** | 0.44 | 0.50 | 0.69 | 0.77 | 0.87 |
| SPP N-SBTDE | 0.55 | 0.68 | 0.76 | **0.51** | 0.80 | 0.94 |
| **Average** | **0.49** | 0.61 | 0.59 | 0.67 | 0.85 | 0.91 |

### A.2 Attention models

We choose wide ResNets with a width of 10 and a depth of 28 as the architecture for the attention models (Wu et al., 2016). All models were trained for 200 epochs with a batch size of 256. In our experiments, the training of all attention models converged before the maximum amount of epochs was reached. We trained all models with an Adam optimizer and multiple learning rates $l \in \{0.0001, 0.001, 0.01\}$ and used the models with the lowest loss achieved in training for the remaining experiments. However, we observed only negligible differences in the different learning rates for all attention models. For all models, we used 10 fixation steps during training and updated the weights for every fixation.

## B   Scanpath Generation - Normalized Results

Throughout the paper unnormalized results are shown for all metrics that compare human and artificial scanpaths. This was done to make the results comparable to other work. In Figure 6 and Table 3 we show normalized results, which are easier to interpret. The metrics are normalized such that the intra-human distance equals 0 and the distance between human and random scanpaths equals 1. More specifically, for each $score(m, k)$ obtained by the model $m$ for sub-sequence length $k$, we consider the normalized score

$$N\text{-}score(m, k) = \frac{score(m, k) - score(Human, k)}{score(Random, k) - score(Human, k)}.$$

We denote the normalized SDE by *N-SDE* and the normalized SBTDE by *N-SBTDE*.

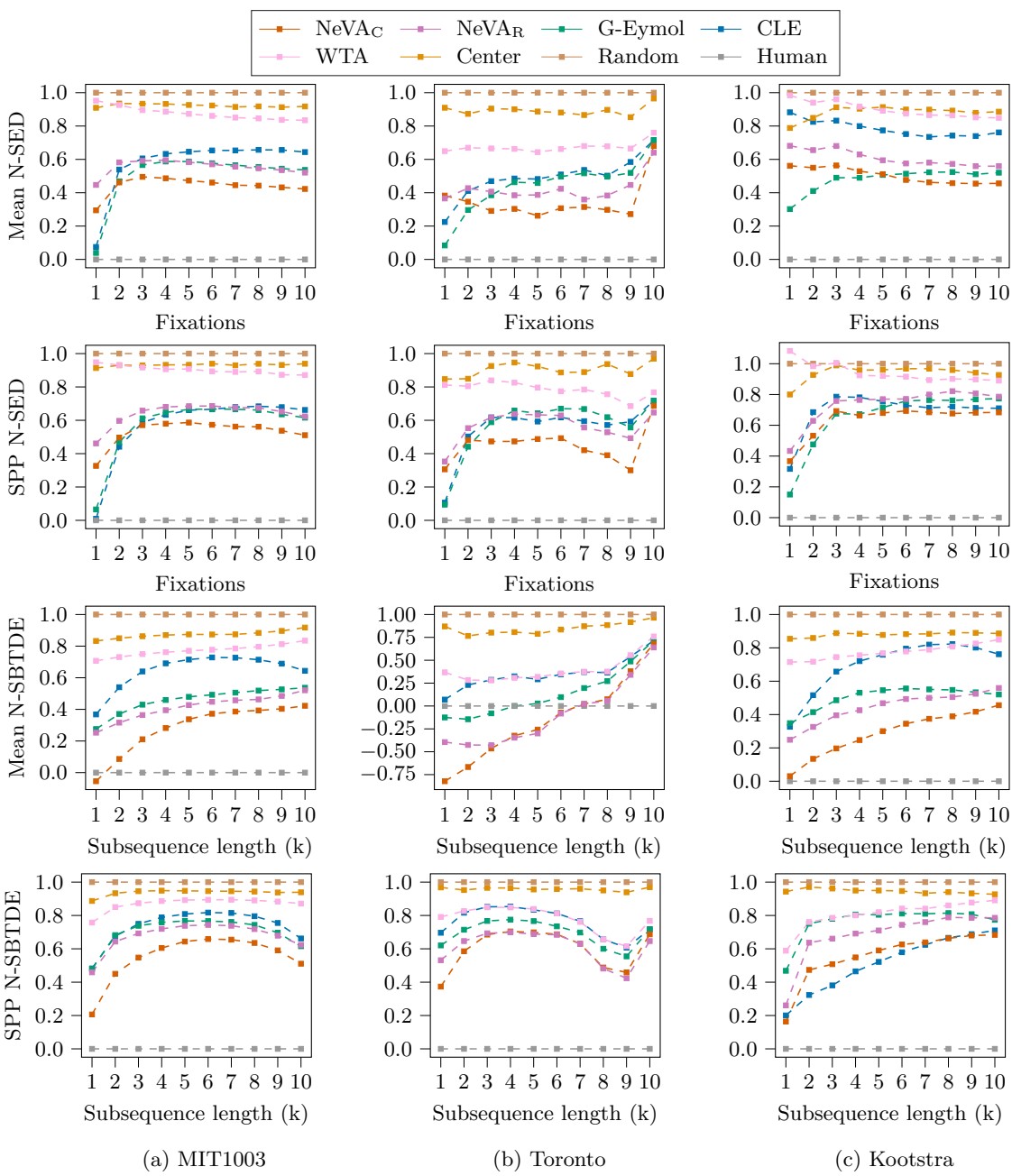

Figure 6: Similarity of human and artificially generated scanpaths from different methods assessed with 4 different distance metrics on 3 datasets. Each row corresponds to a different metric and each column to a different dataset. A **lower** score corresponds to a higher similarity to human scanpaths for each metric. The human baseline is calculated by comparing scanpaths between different subjects and can be considered as a gold standard (metrics are **normalized** such that the average distance is **0** and **1** for humans and random fixations, respectively).

## C Influence of the Forgetting Hyperparameter

We investigated the influence of the forgetting hyperparameter $\gamma$ on the similarity between the generated scanpaths and human scanpaths and the distribution of the saccade amplitudes. Figure 7 summarizes the influence of $\gamma$ on the normalized mean SED. Note that all configurations lead to the same first two fixations as NeVA does only have access to the blurred stimulus for the first fixation and only uses the currently perceived stimulus for the second fixation. Afterward, past information is considered and the value of $\gamma$ has an influence on the generated fixations. For the classification supervised $\text{NeVA}_\text{C}$ attention model the influence of $\gamma$ considerably increases in the later fixations. For the $\text{NeVA}_\text{R}$ configuration this effect is less distinct. Furthermore, the best value for $\gamma$ changes during the scanpath for both configurations. Averaged over all metrics, $\gamma = 0.3$ resulted in the highest similarity between NeVA and human scanpaths in our experiments.

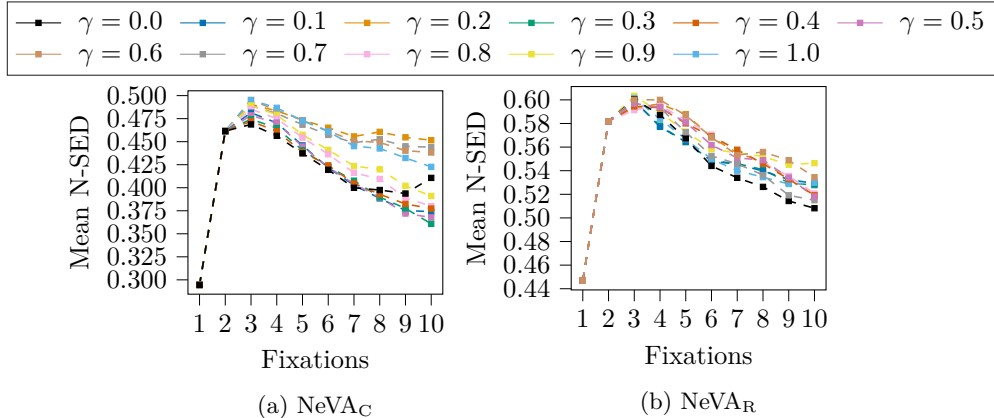

Figure 7: SED of different NeVA configurations for the **MIT1003** dataset. A **lower** score corresponds to a higher similarity to human scanpaths. Metrics are **normalized** such that the average distance is **0** and **1** for humans and random fixations, respectively.

## D NeVA optimization algorithm

Instead of training an attention model to predict scanpaths for a given stimulus NeVA can be implemented as an optimization algorithm. Here, the scanpaths are directly calculated by using the loss of the task model and iteratively updating the attention position for each fixation. The optimization for each attention position $\xi$ can be solved quickly by proposing several possible attention position candidates evenly distributed over the image at each step. For each candidate, the loss value of the task model is then computed to find the best attention position. Given a sufficient number of candidates, this implementation eliminates the need for iterative updates. As a result, only one forward pass is required to compute an attention position for a given task model. Quantitative differences between optimization-based NeVA-O and training-based NeVA are given in Figure 8.

## E Application in downstream tasks

Complementing the results in Table 2, Figure 9 illustrates the accuracy over the number of fixations for the different datasets and attention models.

## F Influence of the predicted label

Figure 10 and Table 4 summarize the influence of the predicted label on the quality of the generated scanpath. In both cases, the label is changed with an adversarial attack on the clean image. Figure 10 illustrates that

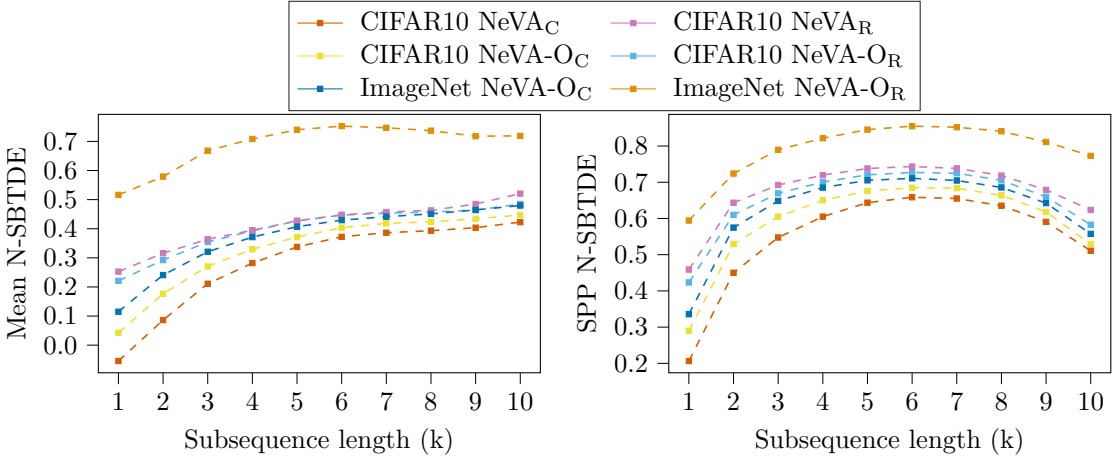

Figure 8: SBTDE distance of human and artificially generated scanpaths of different NeVA configurations for the MIT1003 dataset. A **lower** score corresponds to a higher similarity to human scanpaths. Metrics are **normalized** such that the average distance is **0** and **1** for humans and random fixations, respectively.

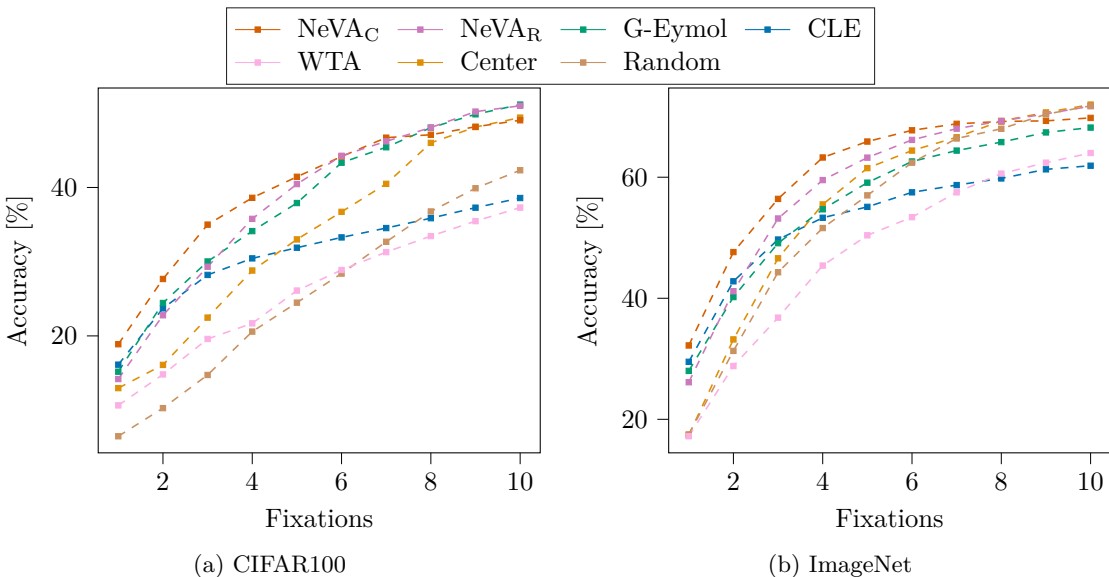

Figure 9: Accuracy of a classification task, where the input is a blurred version of the original input. The input is sequentially deblurred using scanpaths from different methods and the accuracy is shown over the different fixations of the scanpath. Both subsets of the CIFAR100 and the ImageNet dataset (1000 images each) are considered. For the proposed NeVA method C indicates an attention model trained on a classification downstream task and R an attention model trained on a reconstruction downstream task.

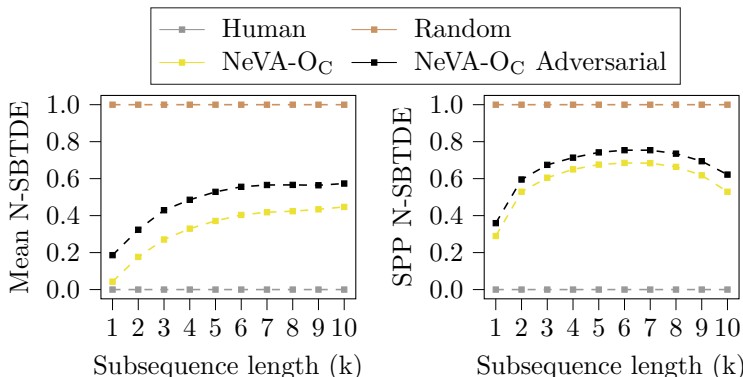

Figure 10: Similarity of human and artificially generated scanpaths for adversarially attacked images and normal images for the **MIT1003** dataset. A **lower** score corresponds to a higher similarity to human scanpaths for the two metrics. The human baseline is calculated by comparing scanpaths between different subjects and can be considered as a gold standard (metrics are **normalized** such that the average distance is **0** and **1** for humans and random fixations, respectively).

Table 4: Accuracy of a classification task, where the input is a blurred version of the original input. The input is sequentially deblurred using scanpaths from the NeVA method and the accuracy is averaged over the scanpath. Both subsets of the CIFAR100 and the ImageNet dataset (1000 images each) are considered. For the column NeVA attacked, the images where attacked by an adversary to change the class label used for supervision during scanpath generation.

| Datasets | NeVA | NeVA Attacked | Center |
|----------|------|---------------|--------|
| CIFAR100 | 35.79 | 30.94 | 33.41 |
| ImageNet | 56.12 | 50.93 | 55.72 |

the similarity to human scanpaths decreases if the label is altered, while Table 4 shows a decline in accuracy in a classification downstream task for the attacked NeVA algorithm.

## G  Saccade amplitude analysis

The scanpath metrics used in the paper do not directly convey information about the nature of similarities and differences between the scanpaths of the different models (e.g., do they generally attend a variety of objects, perform high amplitude saccades etc.). To explore the priors that the different scanpath methods have on the properties of the generated scanpaths, we performed additional analysis to investigate the saccade amplitudes of the generated scanpaths and compared them to the saccade amplitude distribution of humans. The saccade amplitude is defined as the angular distance traveled by the eye during a saccadic movement. To compute it, we convert the distance in pixels to centimeters, by considering resolution and size of the presentation screen used during data collection. Then, taking into account the distance of the subject from the screen, we convert the distance in centimeters to degrees of visual angle by applying trigonometric rules. This analysis is particularly interesting as all methods do not explicitly constrain the scanpaths to have human-like characteristics. Thus, this experiment offers insights into what mechanisms lead to human-like saccades. Figure 11 compares the saccade amplitude of the scanpaths of the different approaches. Scanpaths of trained attention models show higher total saccade amplitudes than scanpaths that are directly optimized with the task-models for both tasks and all datasets. Furthermore, scanpaths generated under reconstruction-task supervision are generally longer than scanpaths generated by classification-task supervision. In addition, we calculated the distribution of saccade amplitudes between different fixations for all models. The results are summarized in Figure 12. Humans exhibit a peak around $2-4$ degrees of visual angle, in accordance with prior work (Tatler et al., 2005). The NeVA configurations also generate saccades in the same range of amplitudes, but clearly show a second peak around the value of 10 degrees. Among competitors, G-Eymol

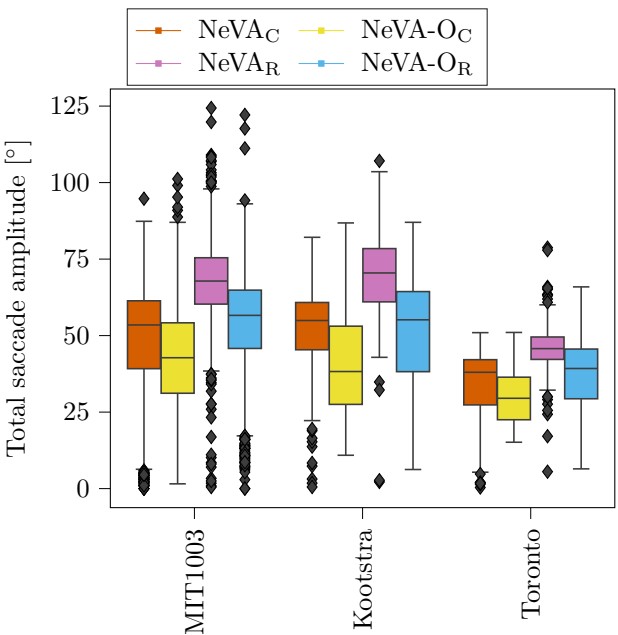

dataset

Figure 11: Boxplots of total saccade amplitudes of different NeVA configurations for the three considered datasets.

exhibits a similar pattern as humans, in accordance with prior results (Zanca et al., 2020b). CLE and WTA algorithms, instead, produce wider saccades, and saccade amplitudes appear to be normally distributed. Random and Center baselines have quite different and unnatural behaviors, with the first one having a high prevalence of shorter saccades.

In Figure 13 the saccade amplitude distributions for different values of $\gamma$ are shown. For both NeVA attention models (trained with classification and reconstruction supervision) lower values of *gamma* lead to a higher similarity to the human saccade amplitude distribution. We observed that setting *gamma* to values close to zero will substantially reduce the saccade amplitude between later fixations. Thus, the number of saccades between 10 and 15 degrees is reduced, which leads to a higher similarity to the human saccade amplitude distribution. We could not see a decrease in saccade amplitude between later fixations for humans with scanpaths of length 10. We will explore this phenomenon in more detail in future work.

Overall, the optimized version of NeVA using classification as a downstream task exhibited the lowest Kullback Leibler (KL) divergence between its distribution of saccade amplitudes compared to those of humans. However, the other approaches also showed a considerably lower KL divergence than the center and random baseline. This result is surprising as none of the approaches contains priors that explicitly induce this behavior. We further did not observe a correlation between the similarity of saccade amplitudes and the other metrics used to assess the similarity of the scanpaths. Both aspect needs to be investigated further in future work, for example by studying the dynamics of saccade amplitudes over time.

## H  Qualitative analysis

Here we report a qualitative comparison between different methods. Figure 14 exemplifies how NeVA improves upon prior approaches. In the first example, the bird is hardly visible against the background.

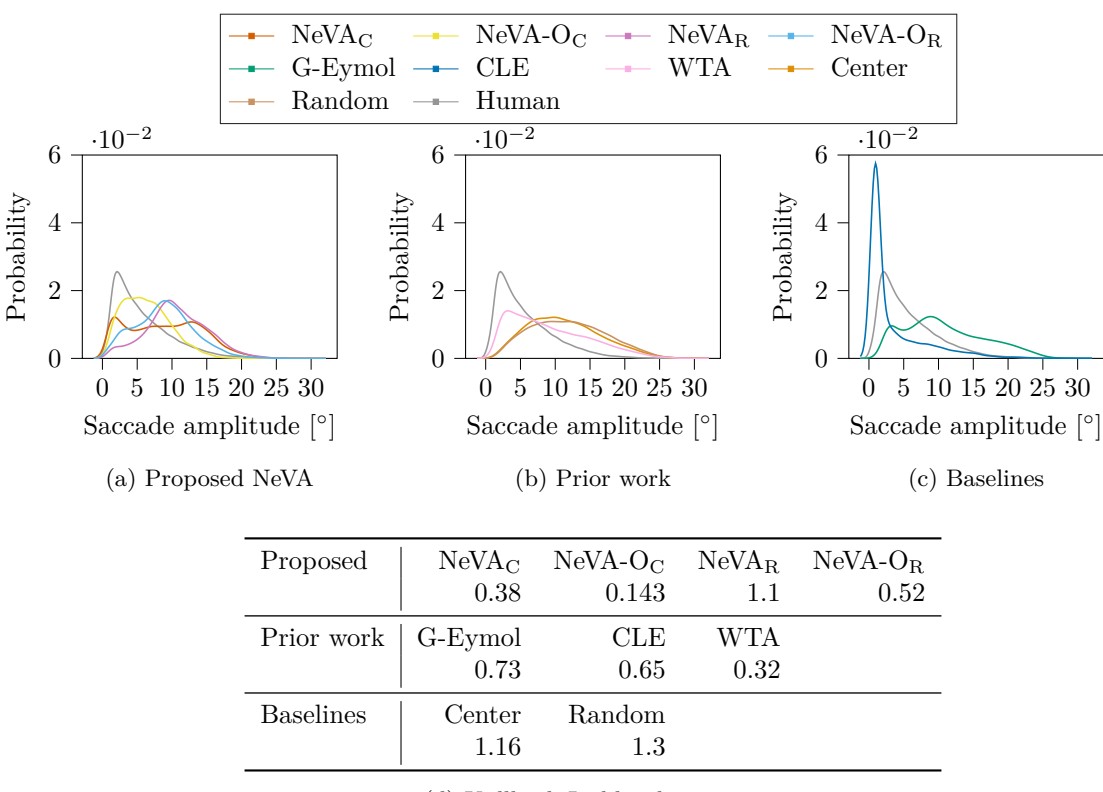

(a) Proposed NeVA      (b) Prior work      (c) Baselines

| Proposed | NeVA$_C$ | NeVA-O$_C$ | NeVA$_R$ | NeVA-O$_R$ |
|---|---|---|---|---|
| | 0.38 | 0.143 | 1.1 | 0.52 |
| Prior work | G-Eymol | CLE | WTA | |
| | 0.73 | 0.65 | 0.32 | |
| Baselines | Center | Random | | |
| | 1.16 | 1.3 | | |

(d) Kullback-Leibler-divergence

Figure 12: (a-c) Saccade amplitude distributions of scanpaths generated by different methods (c) Kullback-Leibler-divergence between the saccade amplitude distributions of scanpaths generated by different methods and the saccade amplitude distribution of human scanpaths. Shown results are averaged over all datasets.

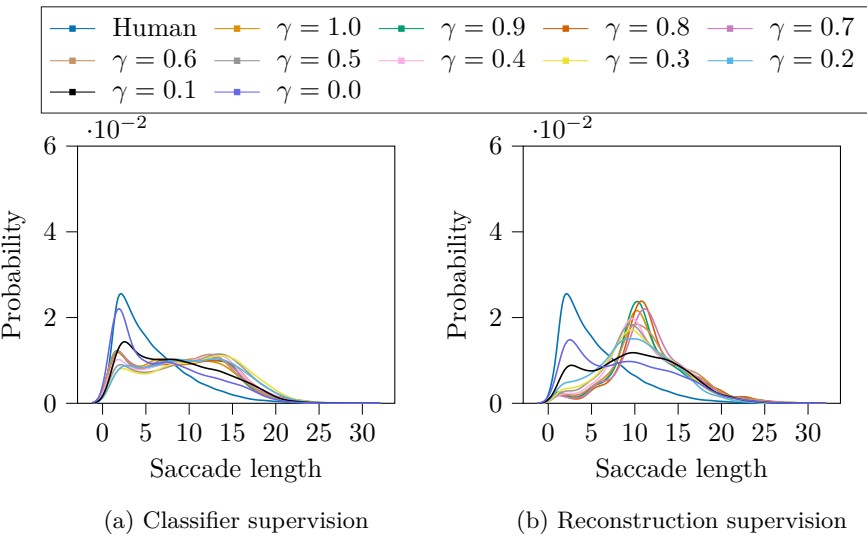

(a) Classifier supervision

(b) Reconstruction supervision

Figure 13: Saccade amplitude distributions of scanpaths generated with different values of the forgetting hyperparameter $\gamma$. Shown results are averaged over all datasets.

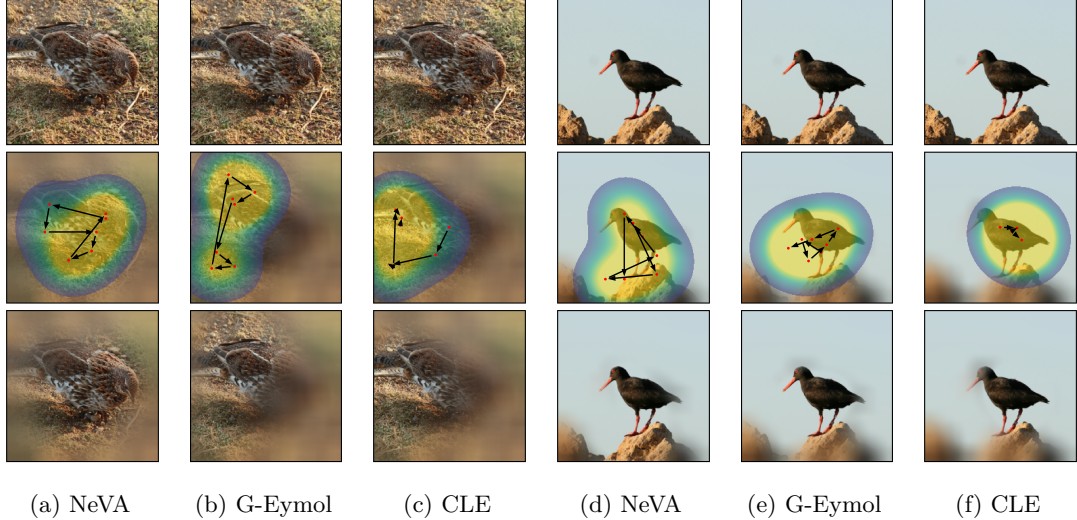

(a) NeVA (b) G-Eymol (c) CLE (d) NeVA (e) G-Eymol (f) CLE

Figure 14: Example scanpaths generated by NeVA and prior approaches. The first row shows the original stimulus, the second row shows the foveation heatmaps and fixations, and the last row shows the internal representation after the last fixation.

Bottom-up approaches thus have a hard time detecting relevant features. In the second example, existing approaches fail to attend the head of the foreground object.

