# OpenReview forum: "Behind the Machine’s Gaze: Neural Networks with Biologically-inspired Constraints Exhibit Human-like Visual Attention"
_TMLR — Accepted by TMLR_

### Review · Reviewer_n3FM · 2022-06-21

**Summary Of Contributions:**

This paper introduces a new model to predict the sequence of human eye-fixations in an image. The model predicts the position of the next eye-fixation by using only the previously foveated image regions, unlike previous works that use the information of the entire image, even from image regions that have not been foveated. Also, the model uses a visual task such as object recognition or image reconstruction as training objective, while previous works used the prediction of the sequence of eye-fixations as the model's objective. In this way, the model introduced in this paper models how top-down information drives the sequence of eye-fixations. Comparisons with several baselines demonstrate the effectiveness of the model.

**Broader Impact Concerns:**

I think there are no broader impact concerns.

**Requested Changes:**

- Positioning of the work: I think the paper would benefit from mentioning what aspects remain not biologically plausible. This could done  by motivating why the paper focuses on the partial information and the top-down signals, and by explaining that there is a number of aspects that remain unexplored and why would be interesting to explore them in the future building from this work.
- Implications of the work: Some discussion of whether something new about human vision has been learned from this work and how it is expected that this model will help understanding human vision in the future or developing new applications.
- Internal representations: A discussion about the biologically plausibility of the internal representation would be beneficial to understand this critical component of the model. It seems to me that the proposed internal representation is not biologically plausible. It would be awesome to see some experiments on different alternatives for this component. For example, what is the relation with [1].
- Partial information vs full: What is the intuition behind the result that shows that a model with partial image information obtains better results than a model with full image information? The latter has access to the same information as the former, so in principle the latter has the capability of making the same predictions as the former. I may be not fully understanding the baselines in section 4.1, and some clarification may be necessary.

-----------

Technical aspects: I think the following points require the attention of the authors:

* Page 3, end of first paragraph: Zhao et al., 2015 introduces a dataset not a model, the right citation is [2].
* Perfect information / Imperfect information: These terms are not precise.
* I think \alpha is never defined
* Is the pre-training of ResNet used in the classification task an advantage over the ResNet trained from scratch in the reconstruction task? Also, is the pre-trained ResNet from RobustBench somehow trained to be robust to adversarial examples, or what is the reason to use RobustBench?
* Loss behavior in section 4.3: would increasing the learning rate improve the convergence speed of the reconstruction task? If so, would the authors expect that the eye-fixation scan-path be more accurate? It seems that the answer would be 'yes' after reading section 4.3, but this is unclear, why would convergence speed be related to final accuracy?
* The accuracy of the best model in ImageNet is 61.03%. Is this top-1? I think some comment comparing this with the accuracy of ResNet would help understanding this result in a broader context.
* In the paragraph before 4.6 there is a typo: "W we"
* Analysis of the scan-paths in the presence of adversarial examples: the paper misses the take-away of the experiment. The paper explains some of the technical details, but I am not sure what is the take-away of the result in Figure 7. I think it would be best to briefly mention the take-away in 4.5.
* I think it would be helpful to provide some comparison with a method that directly trains to predict the sequence of eye-fixations, in order to assess the impact of training the model to classification or reconstruction, specially given that this is one of the main novelties of this work.
* In A.1 it is mentioned that the training is stop when the loss converges, while in A.2 the models are trained for 200 epochs. I think it would be best to explain these differences.

[1] Volodymyr Mnih, Nicolas Heess, Alex Graves, and Koray Kavukcuoglu. Recurrent models of visual attention. In NIPS’14

[2] Xun Huang, Chengyao Shen, Xavier Boix, Qi Zhao, SALICON: Reducing the semantic gap in saliency prediction by adapting deep neural networks. ICCV'15




**Strengths And Weaknesses:**

The paper is strong in the following aspects:
- The model is more biologically plausible than previous models in two key aspects: modeling the availability of visual information available to the model during the sequence of eye-fixations, and also, the eye-fixations are driven by top-down signals related to a visual task.
- Simplicity of the model. While the model is trying to mimic in more detail the human visual system than previous works, the model remains intuitive and allows to be effectively trained.
- Experiments not only try to show improvements from previous works, but also investigate properties of the model such as the effect of the specificity of the label and the task that the model is trained on.

The main weaknesses at a general level are the following (below in "requested changes" I explain them in more detail):
- Positioning of the work: It is unclear why the focus of this work is on the biologically plausibility of the partial information available during the sequence of eye-fixation and the top-down signal that drives them, and not other key aspects of visual attention (eg. eye-fixation duration, inter-subject variability, free viewing vs other tasks, etc.). In other words, the aspects that this work focuses are interesting, but it is not clear why the authors have chosen to study these aspects and not other aspects.
- Implications of the work: While developing accurate models of the brain is important for understanding human vision and for a number of applications, it is unclear what can be derived from this model besides that it works better and that is more biologically plausible than previous models.
- Internal representations: The proposed internal representation to accumulate the observed image scan-path is not biologically plausible. The integration is done at the stimulus level rather than with some form of working memory neural representation. Thus, while the paper makes progress introducing more biologically plausible aspects to the model, it also introduces some components that are not biologically plausible.
- Partial information vs full: It is unclear why the proposed model with partial image information performs better than models with access to the full image.
- Some technical aspects of the experiments are unclear.

---

> ### Author Response · Authors · 2022-06-23
> **Response to reviewer n3FM (1/2)**
>
> We are grateful to the reviewer for the extensive comments. We believe these will improve the final paper, especially in the positioning of the research and the clarity of the supported claims.
>
> ***"Positioning of the work: I think the paper would benefit from mentioning what aspects remain not biologically plausible. [...]"***
>
> In this first work, we focus on foveated vision as it is present at the very first steps of the vision "pipeline". We believe that having limited access to visual input is crucial to the attention mechanism. For the sake of the clear interpretation of the results we have limited ourselves to this component in the first experiments. The reviewer's comment arrives on time and in fact, it is of great interest to extend it to other biological constraints in later stages.
> We added a paragraph where we explain why we specifically focus on the biological constraint of foveated vision. Additionally, we describe the limitations of our approach regarding other biological constraints such as those stated by the reviewer (eye-fixation duration, etc.).
> We clarified that we did not aim to create a biologically plausible model of human attention but rather are interested if biological constraints can help to generate more human-like scanpaths. In this context, we also address why we use top-down signal to drive scanpath generation.
>
> ***"Implications of the work: Some discussion of whether something new about human vision has been learned [...]"***
>
> The key takeaway of this work is that adding biological constraints to neural networks makes them explore images in a more human-like manner. We believe that this finding can be integrated into existing approaches that model human attention. Moreover, it motivates further studies on other biological constraints and if adding them to neural networks further improves the alignment between artificial and human scanpaths.
> We will make our claim clearer and more explicit in the final version of the paper.
>
> ***"Internal representations: A discussion about the biologically plausibility of the internal representation would be beneficial [...]"***
>
> We agree with the reviewer that the proposed model for internal representation may not be biologically plausible (which was beyond our scope of our paper proving). Still, some form of “short-term memory” is necessary for the model to generate useful scanpaths. Without adding this component the attention model will only focus on the position that is most important for classification and will not consider other fixation positions throughout the scanpath.
> We have implemented the simplest version of memory possible, where one parameter controls forgetfulness. Since the mechanism is extremely simple, it is easier to interpret.
> We decided to not model this mechanism as an RNN in this first version for the following reasons:
> 1) A recurrent attention model requires a recurrent task model that can also keep information about past fixation positions. This makes it impossible to use pretrained models and more difficult for other researchers to test NeVA.
> 2) With our simple approach we could directly control the forgetting parameter and assess the influence on the generated scanpaths. While this is also possible with RNNs it would be more complicated to do so.
> We added these explanations to the paper and provide an outlook on how to integrate more sophisticated methods.
> As discussed above, one could explore if adding biological plausibility to this (and other) component as well increases the alignment between artificial and human scanpaths.
>
> ***"Partial information vs full: What is the intuition behind the result that shows that a model with partial image information obtains better results than a model with full image information? [...]"***
>
> We believe that the reduced alignment emerges because humans are not able to predict which next fixation position would be optimal to attend. Rather, we humans have to rely on partial information in the periphery to make an approximated guess. Thus, greedily searching for the “optimal” fixation positions does not lead to human-like behavior. This is supported by our analysis of saccade amplitudes in the appendix. We agree that it would benefit the paper if this is already explored in section 4.1 and not only in the discussion to make it more clear to the reader.

---

> ### Author Response · Authors · 2022-06-23
> **Response to reviewer n3FM (2/2)**
>
> ***"Technical aspects [...]"***
>
> We addressed all technical concerns of the reviewer. The more complex concerns are resolved as follows:
> - We investigated if the trained denoiser is able to reconstruct noisy images in a meaningful way and observed no visual differences between the original images and the reconstructed images. Thus, we do not believe that using a pretrained model would change the results. We did not use a robust task model in our experiments. We used RobustBench for convenience since it contains sev-eral different models.
> - The loss behavior is shown for the inference part of the algorithm and does not depend on the learning rate. The plot shows how the loss of the task model changes for a scanpath generated by the attention model. Since more parts of the image are revealed over time the loss decreases. We do not expect a decrease in loss to necessarily correspond to more human-like scanpaths. Other-wise, the additionally optimization-based approach would generate more human-like scanpaths. We added further discussion about this experimentto the paper.
> - The adversarial experiments demonstrate that the label has a considerable influence on the scan-path even if the predicted label is not the same as the one in the image (this is the case for most of the MIT1003 images since the image labels do not overlap with CIFR10). This indicates that the model encodes label-specific concepts in its predictions which affect the exploration in a substan-tial manner even for unknown objects. We added a takeaway to the relevant section.
> - One of the goals of this work was to investigate which tasks are most likely to drive human atten-tion. Training an algorithm in a supervised way to predict scanpaths makes it difficult to explain which features/mechanism the trained model uses to generate the scanpaths. Creating the best possible model of human attention is not the aim of our work. We made this more clear in the pa-per.

---

> > ### Comment · Reviewer_n3FM · 2022-09-17
> > **Changes in the paper**
> >
> > I can not find in the paper where are the changes made for each of the comments. Please indicate page and paragraph where each change has been made to address the different comments.
> >
> > It is not clear what the following claims means:
> >
> > "We clarified that we did not aim to create a biologically plausible model of human attention but rather are interested if biological constraints can help to generate more human-like scanpaths."
> > 1. What is the difference between making a model more biologically plausible and adding biological constraints?
> > 2. Can you explain why making a model that generates more human-like scanpaths is not making the model more biologically plausible?
> >
> > "The key takeaway of this work is that adding biological constraints to neural networks makes them explore images in a more human-like manner. ": The paper also adds a number of other constraints that are not biologically motivated, eg. the memory. What evidence is provided that guarantees that the improvements come from the new biological constraints and not from the newly introduced non biological constraints?

---

> > > ### Author Response · Authors · 2022-09-19
> > > **Clarifications**
> > >
> > > ***I can not find in the paper where are the changes made for each of the comments. Please indicate page and paragraph where each change has been made to address the different comments.***
> > >
> > > Here’s the detailed list of changes made on the first revision:
> > > - Introduction, last paragraph before the bullet point list. We improve the positioning of the paper.
> > > - Discussion, last paragraph. We discuss study implications and limitations.
> > > - Section 3.2.3, before the last paragraph. We discuss our simplistic choice for internal representations.
> > > - Section 4.1. We now refer the reader to section 5 for explanations regarding partial-vs-full approaches.
> > > - Section 4.4 now contains more information about the loss and accuracy behavior for downstream tasks.
> > > - Section 4.5, last paragraph. Takeaway message on the influence of the predicted label.
> > >
> > >
> > > ***"We clarified that we did not aim to create a biologically plausible model of human attention but rather are interested if biological constraints can help to generate more human-like scanpaths."
> > > What is the difference between making a model more biologically plausible and adding biological constraints?
> > > Can you explain why making a model that generates more human-like scanpaths is not making the model more biologically plausible?***
> > >
> > > Thank you for pointing out that our wording was somewhat misleading. We believe that these constraints make the model more biologically plausible. With this sentence, we wanted to clarify that we do not claim that the whole model is biologically plausible (i.e., the inner workings of the neural network). Due to the complexity of the problem, we restricted our analysis to the perception part (i.e., foveated vision) and the underlying task, and investigated their impact on the generated scanpaths. We believe that adding more biological constraints will further improve the alignment between artificial and human scanpaths, which needs to be explored in future work.
> > >
> > > ***"The key takeaway of this work is that adding biological constraints to neural networks makes them explore images in a more human-like manner. ": The paper also adds a number of other constraints that are not biologically motivated, eg. the memory. What evidence is provided that guarantees that the improvements come from the new biological constraints and not from the newly introduced non-biological constraints?***
> > >
> > > For sequential modeling, previous fixations must be kept in memory. For the sake of clarity of the results, we decided to cumulate the fixations with the simplest possible approach, i.e., a weighted sum. We discussed this in section 3.2.3. We believe that our ablation study analyzing different values with the forgetting factor (Appendix C) can well explain the contribution of internal representation (or, memory) in its current implementation to the generated scanpaths.

---

> > > > ### Comment · Reviewer_n3FM · 2022-09-19
> > > > **Unclear claims**
> > > >
> > > > I am having a hard time trying to understand what is the goal of this paper and what is trying to claim. When the paper is updated, for each reviewer concern please indicate what changes in the paper have been made (including location in the text).

---

### Review · Reviewer_qd44 · 2022-07-06

**Summary Of Contributions:**

The paper presents a method to generate visual scanpaths in a top-down matter which is focus on incorporating the signal from high-level visual tasks such as image classification and reconstruction.

Contributions:
* Incorporate the task guidance into the attentional modeling
* Use partial information (e.g., foveated) to generate scanpaths
* Extension of existing metrics to account for stochastic components in human behaviour

**Broader Impact Concerns:**

No concerns about broader impact.

**Requested Changes:**

Clarity of the paper
* Abstract. “[…] without incorporating the signal of a high-level visual tasks that have shown to partially guide human attention.“ I’d suggest to add some motivation about why it’s important to have visual tasks to guide attention. It’s written in the introduction already, but it’s also important to have a sentence here in the abstract too.
* Introduction. “When predicting saliency, the temporal dimension is lost, and the scanpath generation problem still remains unsolved.” It’s not clear why temporal order is important for this task. One could consider the points of scan as independent points and compute saliency maps. Could you please provide an example to motivate this claim?
* Fig 1 in page 2 is distracting, when reading the introduction. Please move after Sec. 2.2.
* Sec. 2.2.3, first equation. It’s a bit hard to imagine the effect of this equation. I’d suggest to add a short description of what intuitively is happening and what is the outcome of such operation.
* Page 5. “the categorical-cross-entropy loss for the classification task and the mean-squared error loss for the reconstruction task.” Please be clear that these are two different networks, otherwise the reader can thing that you trained for both combined loss.
* Sec. 3.3.2. “To take into account the stochastic component of the attention process” I suggest the authors to expand a bit the meaning of this sentence. I know that there is a reference, but it’d be great to have a sentence describing what stochastic complement means.
* Sec. 3.3.2. “to make them robust with respect to changes in stimulus resolution and scanpath length” To me it’s not clear how the proposed metric is more robust wrt (Wang et al., 2011; Zanca et al., 2019b). Can you provide more insight and maybe an example?

Technical correctness
* Introduction. The authors claim 2 limitations of current approaches. It would be interesting to see a comparison with some of existing works that incorporate task guidance and use partial information (foveation). Some examples are [A1, A2], but there might be others that are more recent.
* Second equation in page 5 and last of page 6. What’s \alpha? Is it the agent a? If so, please fix notation. Or clarify it’s definition.
* Sec 2.2.2. The method at every step t works on the full smoothed image. However in practice, humans do not have access to the “full image” when looking at something (given the limited field of view). I’m wondering if it would be possible to extend the method to deal with sub-regions of the image instead of full images. This is also important from the computational perspective, because now the method have less pixels to compute at each time t. I’d like to either see a discussion about this and/or an experiment showing that the method works under these more realistic conditions.
* Page 7. It’s quite difficult to understand how the attention model is defined and how the next fixation is generated. Is there a way to expand this part and have it in math form? Alternatively, it would be appreciated if the authors try to explain what model it is and how the outputs are generated for a given input. It’s also confusing because training and prediction part are mixed. So it’s unclear what happens in the two cases.

[A1] Larochelle, Hugo, and Geoffrey E. Hinton. "Learning to combine foveal glimpses with a third-order Boltzmann machine." Advances in neural information processing systems 23 (2010).

[A2] Denil, Misha, et al. "Learning where to attend with deep architectures for image tracking." Neural computation 24.8 (2012): 2151-2184.

Experiments
* Page 8. “we use a wide ResNet that was trained on the CIFAR10 dataset“ Isn’t  this already at very low resolution (32x32)? What is the effect of blurring if the images are already quite low resolution?
* Sec. 4.1. It’s unclear why the CIFAR10 models are much better than ImageNet models. It makes sense to me that higher resolution models should be more beneficial to capture details in the fovea, when compared to the low resolution of CIFAR10. Maybe the ImageNet model did not converge well during training? Or maybe there is a big domain shift between datasets? I’d like to see an argumentation about this point.
* Sec. 4.2. “We only consider the first 10 fixations of human scanpaths and discard recordings with less than 10 fixations”. Why can’t we use this data with less fixations? One can run the evaluation by varying the length of fixations in the test set (e.g., only data with 1, with 2, …). This would give us a better understanding what happens with examples with shorter scanpaths too.
* Sec. 4.2 and Table 1. “To simplify the interpretation of the results all metrics are normalized such that the intra-human distance equals 0 and the distance between human and random scanpaths equals 1.” I’d really suggest the authors to provide absolute numbers instead of relative. This is to encourage comparisons with future paper and make it easier.
* Table 2. Even if it is not very relevant to this experiment, I’d suggest to add the best accuracies on the datasets of the current state of the art methods. It’s not for direct comparison, but to show the reader where we are with attentional methods compared the best results for the task.
* Fig. 4. It’d be interested to see the qualitative results of NeVA scanpaths compared to the second best performing method.

**Strengths And Weaknesses:**

Strengths
* The introduction, motivations and discussions of the work are well written.
* The proposed model seems to be sound and correct (apart some clarifications to be dealt with).
* In-depth experimental analysis, comparing with other methods for different datasets and evaluation metrics.

Weaknesses
* Almost all reported results are in relative format. To encourage other scientists to use this method, I'd suggest to use absolute numbers.
* The model might be computationally expensive, since the modified version of the full image is processed multiple times (as many times as fixations).
* Some parts of the paper are hard to follow (especially how the new fixation point is generated).
* Results using ImageNet models are not convincing.

Please see "requested changes" for more information about how to improve the weaknesses.

---

> ### Author Response · Authors · 2022-07-08
> **Response to Reviewer qd44 (1/2)**
>
> We are grateful to the reviewer for the extensive comments and detailed analysis of the paper. Please find below a detailed point-by-point response to all comments.
>
> ***WEAKNESSES***
>
> ***“Almost all reported results are in relative format. […] I'd suggest to use absolute numbers.“***
>
> We added the non-normalized results to the supplement.
>
> ***“The model might be computationally expensive, since the modified version of the full image is processed multiple times (as many times as fixations).“***
>
> Sequentiality is inherent to the problem formulation, especially when one enforces foveated vision. One could also imagine a system that outputs a scanpath in one step, but this is biologically not plausible. Still, in our experiments, we apply the CIFAR10-based model to a lower resolution version of the image (32x32) which greatly reduced the computational overhead for datasets like ImageNet.
>
> ***“Some parts of the paper are hard to follow (especially how the new fixation point is generated).“***
>
> We provided additional explanations of formulas where possible. (see also “Requested changes”).
>
> ***“Results using ImageNet models are not convincing.“***
>
> We extended the explanations of the ImageNet results in the paper (see also “Requested changes”).
>
>
> ***REQUESTED CHANGES***
>
>
> ***CLARITY***
>
> ***“Abstract […] I’d suggest to add some motivation about why it’s important to have visual tasks to guide attention. […]“***
>
> We have added additional explanations to the abstract in order to clarify how a mechanism capable of incorporating a task signal "flexibly provides top-down guidance to the scanpath". The reader will find more in depth explanations in the main text.
>
> ***“Introduction. […] It’s not clear why temporal order is important for this task. […]“***
>
> We agree that saliency can be predicted by accumulating saliency maps of individual fixations over a scanpath. We reformulated this sentence to highlight that it’s not possible to generate a scanpath from a single saliency map as the order of the fixations is not directly encoded (i.e., two scanpaths that are the reverse of each other would generate the same saliency map depending on the algorithm used).
>
> ***“Fig 1 in page 2 is distracting […]“***
>
> We solved this drawback by dividing the image into two independent figures. This additionally improves the logical distribution of the images in the manuscript.
>
> ***“Sec. 2.2.3, first equation. It’s a bit hard to imagine the effect of this equation. I’d suggest to add a short description […]“***
>
> While keeping the formulation, we added an intuitive explanation of the internal representation mechanism to explain that it simply cumulates successive foveations, with a discount term \gamma for older fixations.
>
> ***“Page 5. […] Please be clear that these are two different networks, otherwise the reader can thing that you trained for both combined loss.”***
>
> We made it clear that these refer to two different networks.
>
> ***“Sec. 3.3.2. “To take into account the stochastic component of the attention process” I suggest the authors to expand a bit the meaning of this sentence […]”***
>
> We extended the explanation to incorporate that “Previous research suggested as human responses to stimuli are not deterministic and people generally attend to different locations on the same stimulus.
>
> ***“Sec. 3.3.2. “to make them robust with respect to changes in stimulus resolution and scanpath length” To me it’s not clear […]”***
>
> Previous studies show that string-based metrics for scanpath similarity tend to be robust to image resolution (provided the dictionary length is fixed for all stimuli). By computing time delay embeddings, we can normalize the metrics with respect to the scanpath length. Our intuition with the string-based time delay embedding is that, by computing TDE in the string domain, we get both benefits. We explained it more extensively in the referred section.

---

> > ### Comment · Reviewer_qd44 · 2022-07-25
> > **Where is the reviewed version**
> >
> > Unfortunately I do not see a reviewed version of the manuscript or supplementary material. The last modification I see is 25 May 2022. So I'm currently blocked in proceeding with the review.

---

> > > ### Author Response · Authors · 2022-07-25
> > > **Uploaded revision**
> > >
> > > We again thank the reviewers for their helpful comments. We initially planned to submit only one revision to save the reviewers from reading the manuscript multiple times. As requested, we updated our initial version with the proposed changes.

---

> > ### Comment · Reviewer_qd44 · 2022-08-01
> > **Refinements**
> >
> > I want to thank the authors for answering to my comments. The paper improved from the first draft.
> > There are few refinements that I'd like to discuss.
> >
> > **“Introduction. […] It’s not clear why temporal order is important for this task. […]“**
> >
> > Please expand and specifically add a sentence about why encoding the temporal aspect is important. It's important to motivate your work. (this was requested in my previous review)
> >
> >
> > **"Sec. 4.1. It’s unclear why the CIFAR10 models are much better than ImageNet models. […]“**
> >
> > I partially understand the answer from the authors. Why this ("ImageNet-based classifier generally lead to very specific and circumscribed explorations") does not apply to pretrained CIFAR too?
> >
> >
> > **“Sec. 4.2. […] Why can’t we use this data with less fixations?“**
> >
> > It's not about reporting results. This is a recommendation of analyzing in more depth the results for different length of scanpaths. Maybe short paths are more effective and optimized for the task (because subjects watch things in different ways). Also, can the model deal effectively with scanpaths of different length? If so, how does it behave? When does it fail? Maybe including shorter lengths can even help to improve results overall, because you'd have more scanpaths.
> >
> >
> > **“Almost all reported results are in relative format. […] I'd suggest to use absolute numbers.“**
> >
> > I'd still suggest to convert figures and tables of the main paper to absolute numbers. Figures and tables with relative numbers can be always added in the supplementary material.

---

> > > ### Author Response · Authors · 2022-08-01
> > > **Answer to Refinements**
> > >
> > > We are grateful to the reviewer for additional comments that improve the quality of the paper. A new revision has just been uploaded. Please find point-by-point answers below.
> > >
> > > REFINEMENTS
> > >
> > > ***Please expand and specifically add a sentence about why encoding the temporal aspect is important. It's important to motivate your work. (this was requested in my previous review)***
> > >
> > > We provided more extended motivation about the importance of predicting scanpaths, compared to the classical approach of saliency prediction, in the introduction.
> > >
> > > ***I partially understand the answer from the authors. Why this ("ImageNet-based classifier generally lead to very specific and circumscribed explorations") does not apply to pretrained CIFAR too?***
> > >
> > > An ImageNet classifier needs to learn to distinguish $1000$ different classes, which partly share semantic similarities (e.g., multiple breeds of dogs). We argue that the classifier thus learns to focus on descriptive features. In contrast, CIFAR10-based classifiers can learn more high-level features as the classes in CIFAR10 are semantically more different. We extended the explanation in the paper.
> > >
> > > ***It's not about reporting results. This is a recommendation of analyzing in more depth the results for different length of scanpaths. [...]***
> > >
> > > In the current experiments, we already analyse scanpaths at different lengths (see, e.g., fig. 2). This provides insights on the predictive power of different models in the short- and long-term. For example, in line with literature about bottom-up attention guidance, we noticed that NeVA is often worse than bottom-up approaches in the very first fixations (see Discussion, first paragraph). By discarding a small portion of the data (scanpaths shorter than 10 fixations) we make analysis more consistent, since metrics at different lengths are computed for the same explorations. We now specified that “we investigate the behaviour of early fixations by analysing the similarity of scanpaths over time” in the results section.
> > >
> > > ***I'd still suggest to convert figures and tables of the main paper to absolute numbers. Figures and tables with relative numbers can be always added in the supplementary material.***
> > >
> > > We agree with the reviewer and consider the non-normalized results to the main paper.

---

> > > > ### Comment · Reviewer_qd44 · 2022-08-12
> > > > **Done**
> > > >
> > > > No more comments from my side. Thanks for the revisions.

---

> ### Author Response · Authors · 2022-07-08
> **Response to Reviewer qd44 (2/2)**
>
> ***TECHNICAL CORRECTNESS***
>
> ***”Introduction. The authors claim 2 limitations of current approaches. It would be interesting to see a comparison with some of existing works [A1, A2] […]”***
>
> We thank the reviewer for pointing us towards this extremely relevant literature. The referred systems present numerous parallels with our system, even though they were developed for other purposes. We have added them as related works, allowing us to emphasize the practical usefulness of these studies.
>
> ***”Second equation in page 5 and last of page 6. What’s \alpha? […]“***
>
> We fixed the notation.
>
> ***”Sec 2.2.2. The method at every step t works on the full smoothed image. However in practice, humans do not have access to the “full image” when looking at something (given the limited field of view). […]“***
>
> For the considered datasets, during the data collection process, the full stimulus was always in the field of view of the participants, even when they looked at the very edge of it. Thus, we did not need to consider limits of the field of view within our method. Future experiments that work with 360° data should include the field of view in the analysis and algorithms.
>
> ***”Page 7. It’s quite difficult to understand how the attention model is defined and how the next fixation is generated. […]“***
>
> We extended the explanations with a concrete example for training and inference to make it easier to understand for the reader.
>
> ***EXPERIMENTS***
>
> ***“Page 8. […] What is the effect of blurring if the images are already quite low resolution? “***
>
> We saw a considerable decrease in accuracy for the blurred images in the classification task model. Thus, we assume that the influence of blurring is considerable. We added quantitative results to the paper.
>
> ***“Sec. 4.1. It’s unclear why the CIFAR10 models are much better than ImageNet models. […]“***
>
> We compared the optimized version of NeVA for pretrained CIFAR10 and ImageNe models. There was no training involved.  Our explanation for the poor performance with Imagenet trained models are discussed in the last point of the discussion section and visually supported by figure 4: ImageNet-based classifier generally lead to very specific and circumscribed explorations (which are uncommon in humans instructed for free-viewing. We proposed that a better choice might be to consider high-resolution models like YOLO (Redmon et al., 2016) that can detect multiple objects at the same time,  multi-label classification tasks,  or multi-tasking.
>
> ***“Sec. 4.2. […] Why can’t we use this data with less fixations?“***
> Analyzing scanpaths of different lengths would increase the complexity of reporting results considerably. For example, the illustrations in Figure 2 does only work if the scanpaths are all the same length. We chose to report the results in this manner for simplicity and to not shift the focus of the paper.
>
> ***“Table 2. Even if it is not very relevant to this experiment, I’d suggest to add the best accuracies on the datasets of the current state of the art methods. “***
>
> We added this information for completeness.
>
> ***“Fig. 4. It’d be interested to see the qualitative results of NeVA scanpaths compared to the second best performing method.“***
>
> We added a figure with some explanatory examples and qualitative explanations.

---

### Review · Reviewer_uam2 · 2022-09-01

**Summary Of Contributions:**

The paper presented a computational model of top down visual attention, realized as a deep neural network. Inspired by the human vision system, the proposed model – Neural Visual Attention (NVA), selectively attends to a sequence of foveated, local image regions, and extracts information necessary for a high level task (e.g. recognition or reconstruction). In doing so, NVA produces a scanpath of an input image guided by the target task, without using the supervision of eye tracking data. To evaluate the resulting scanpaths, the authors also presented a new metric called String-based time-delay embedding (SBTDE), extending the string editing distance and accounting for scanpaths with varying lengths. The proposed method was evaluated on three public image datasets, where the model generated scanpaths are compared against eye tracking data. The generated scanpaths can be further used for visual recognition and demonstrate promising results.


**Broader Impact Concerns:**

There is no ethical concerns for this paper.

**Requested Changes:**

The main requested changes include
* It is suggested to draw a clear boundary to Mnih et al. and compare the results to Mnih et al. in experiments.
* Please discuss prior works in computer vision
* It will be great to evaluate the proposed model on eye tracking datasets where the subjects are performing a set of tasks.
* Please clarify the visual recognition experiments and justify the performance drop.

Here are some minor comments regarding the presentation and organization of the paper.
* Abstract: Extensive experiments show that outperforms … -> Extensive experiments show that our method outperforms …
* Intro (Page 3 second paragraph): , tend to overfit … ->, they tend to overfit …
* Intro (Page 3 second paragraph): , we focus on unsupervised ->. We focus on …
* Intro (Page 3 last paragraph): Differently by previous proposal -> Different from previous proposals, …
* Intro (Page 3 last paragraph): to expands the action space -> to expand the action space
* Intro: It is suggested to separate out the introduction and the related work section.


**Strengths And Weaknesses:**

**Strength**

The paper addresses a very interesting problem of task guided visual attention, which links scanning patterns of foveated regions to a high level task such as image recognition or reconstruction. The experiments are extensive and the results are encouraging. Further, the paper is reasonably well written, with key ideas and technical components well illustrated.

**Weakness**

My main concern is the similarity to a prior work from Mnih et al., which is unfortunately not discussed in the paper. Mnih et al. also presented a visual attention model that (1) considers task guidance; and (2) uses foveated local regions (i.e., imperfect information as described in this paper). More specifically, their model builds a recurrent neural network that learns to attend to a sequence of foveated local regions guided by the task of image recognition, where visual attention can be learned without eye tracking data. While their paper focused on recognition, their model is very similar (both conceptually and operationally) to the proposed method. For example, their method also generates scanpaths that can be compared against eye tracking data. It remains unclear to me how the proposed method is different to Mnih et al., diminishing the key technical contribution of this paper.


Indeed, I have to point out that the paper missed several related works in computer vision. The authors motivated the work by pointing out two major limitations of existing models on visual attention: (1) the lack of top-down task guidance; and (2) the assumption of full resolution image inputs (i.e., perfect information). However, both have been addressed in prior works. For example, Gao et al. presented a top down visual attention model guided by scene recognition. Mnih et al. and Xu et al. presented models that attend to a series of foveated local regions for image recognition / image captioning. Again, it is not entirely clear how the proposed method adds value to those existing approaches.


While the experiments are extensive, the design does not seem to align with the motivation of the paper. And some of the results are confusing.
* The main experiments are carried out using eye tracking data during free-viewing of images. Task-guided and free-viewing scanpaths are known to have different characteristics. Free-viewing scanpaths can focus on different regions of the same image, and have a much larger variance across subjects, as also recognized by the authors.  Comparing task guided scanpaths (e.g., recognizing objects / scenes) from a model to free-viewing scanpaths from humans seems rather problematic. A higher similarity in the scanpaths does not necessarily suggest the effectiveness of the method. For example, in one of the experiments (Figure 2 row 3 column 2), the model generated scanpaths show higher similarity to humans than other human scanpaths. It seems to me that these evaluations are insufficient to justify the proposed method.
* The visual recognition results are confusing. The text suggested that “the downstream task model is trained on CIFAR10” and “Both (ImageNet and CIFAR100) are not seen by the task model." It seems that the task / NVA model has to be fine-tuned on ImageNet and CIFAR100, as the object categories are different and the image resolutions vary (between CIFAR and ImageNet). However, there is no clear description of the fine-tuning process. Further, the drastic performance drop between (a) using full resolution image and (2) using a sequence of foveated local regions is not well justified. As NVA will deblur the image along the scanpaths, with sufficient fixations, the model could see all pixels in the full resolution image. Further, if the scanpaths did capture the main object in the first few fixations, the model might as well make a correct prediction. It is not clear why such a performance drop still exists. It will be interesting to report the recognition accuracy as a function of the number of fixations.

*References*
* Gao, D., Han, S., & Vasconcelos, N. (2009). Discriminant saliency, the detection of suspicious coincidences, and applications to visual recognition. IEEE Transactions on Pattern Analysis and Machine Intelligence, 31(6), 989-1005.
* Mnih, V., Heess, N., & Graves, A. (2014). Recurrent models of visual attention. Advances in neural information processing systems, 27.
* Xu, K., Ba, J., Kiros, R., Cho, K., Courville, A., Salakhudinov, R., ... & Bengio, Y. (2015, June). Show, attend and tell: Neural image caption generation with visual attention. In International conference on machine learning (pp. 2048-2057). PMLR.

---

### Comment · Reviewer_n3FM · 2022-09-13
**confused with revisions**

Dear authors and editor,

Is there a version of the paper with the changes highlighted in some way that reviewers can access? I am not sure whether this is already available or not. I can see a more recent version of the paper but I can't find the version with the author's changes highlighted.

---

> ### Author Response · Authors · 2022-09-14
> **Revisions summary**
>
> Thanks for the request,
>
> Previously, openreview offered a tool for displaying differences between PDFs, which no longer seems to be available. The AE will probably be able to suggest to us the best way to share changes between versions, in respect of the double-blind process.
>
> In the meantime, we can suggest the alternative solution of using similar services for pdf-diff, such as https://draftable.com/compare, and compare the original submission with subsequent revisions.
>
> For the convenience of the reviewers, we provide a summary of the revisions:
> - 10 May, 2022. First submission.
> - 25 July, 2022. Changes in response to reviewers [n3FM] and [qd44] comments.
> - 1 August, 2022. Minor changes suggested by reviewer [qd44].
> - 2 September, 2022. Changes in response to reviewers [uam2] comments.
> - 14 September, 2022. Minor changes suggested by reviewer [uam2].

---

### Decision · Action_Editors · 2022-10-21

**Recommendation:** Accept with minor revision

**Comment:**

Official recommendations are split. Reviewer n3FM remains unclear about the central claims of the work and recommends rejection. Reviewer qd44 leans towards accept but recognizes n3FM's concerns -- especially regarding the positioning of the work. I agree the positioning could be stronger, but do not think it constitutes a unsubstantiated claim. It is the AE's opinion that the primary claims are clearly stated and sufficiently supported by experiments and that the issues with positioning could be addressed with a minor revision.

For me to recommend acceptance, the claims on biologic plausibility or the model having a "biological constraint" will need to be weakened to reflect that these are not claims about the inner workings of biological systems or that the proposed methods are apt analogs to biological systems. I point authors to the last paragraph of the claims section for further advice on the requested changes.

**Audience:**

The AE believe some of the TMLR audience will be interested in the findings. Unsupervised (in terms of scanpaths) prediction of human attention patterns is a topic of interest within the community. This is evidenced by prior work on the subject and multiple datasets to support its evaluation. Using seemingly unrelated task-specific ML models to derive scan path prediction models is novel in this space and the outcome having greater alignment with human scanpaths is unexpected.

**Claims And Evidence:**

Addressing each claimed primary contributions in the introduction:

(1) "We use common metrics to measure similarity between simulated and human scanpaths, and introduce and motivate a new metric, the String-Based Time-Delay Embeddings (SBTDE), to better account for stochastic components in human behavior."

> This is true to the AE's knowledge and not questioned by the reviewers. While string-edit distance and time-delay embeddings have both been used to evaluate scan paths (as the authors note), the combination appears novel. The arguments for this method are short and not well supported by example or experimentation to demonstrate their value. Future revisions would do well to focus on these arguments if this is a significant contribution.


(2) "In an extensive evaluation with multiple well-established eye-tracking datasets, we show that scanpaths generated by NeVA exhibit the highest similarity to humans compared to state-of-the-art unsupervised methods, as measured by several metrics."

> This is clear and well-supported. One reviewer (uam2) raised concern about using free-viewing scanpaths for evaluation, but the author response seems satisfactory. The proposed model for glimpsing with foveated vision produces scanpaths better matching human scanpaths compared to other unsupervised methods that do not utilize ground-truth scan paths during training. The proposed model itself bares similarity to some prior work in recognition (Mnih et al.) which proposed an integrated sequential glimpsing model for image recognition. The proposed model however relies on pre-trained models to learn glimpsing patterns and the main experimental focus of this paper is the similarity of the scan paths to human behaviors rather than the impact on visual recognition.


(3) "The flexibility of the proposed approach allows us to quantify the contribution of different tasks (classification or reconstruction) to the generated scanpaths, and the importance of partial vision in generating plausible scanpaths (by comparing NeVA with its optimization-based version which assumes perfect vision)."

> This first part of this claim is well supported. The flexibility of using pretrained model gradients does allow this and it is demonstrated in this work. The second portion of this claim is assessed in 4.1 where the proposed approach with partial information (sequence of foveated glimpses) is compared to an optimization approach that selects "ideal" scanpaths based on the entire image. The findings suggest the partial information model (NeVA) outperforms the perfect vision (NeVA-O) in terms of human scanpath similarity. The optimization technique is a confounder in this experiment and a more appropriate baseline would have been a NeVA model that considers the glimpse history as well as the full un-blurred image in making scan path predictions. Such a model could have been trained in the same way as NeVA.


(4) "Lastly, we propose a novel experiment where the utility of scanpaths to solve a downstream visual task on unseen data under the constrained of imperfect information is assessed. In contrast to prior results that mainly focus on similarity to human scanpaths, this experiment aims to asses the usability of the simulated scanpaths for practical applications. Here, we show that NeVA-generated scanpaths are more effective at solving downstream visual tasks on unseen datasets, opening up possibilities for using NeVA models in practical applications where agents have to operate with constrained resources."

> The described experiment does demonstrate that scanpath generated from CIFAR10 trained NeVA models are more effective for new datasets / models than prior work in generating scan paths. That said, it is not clear how to interpret the significance of this results -- either to practice or to our understanding of these systems.

Outside of the claims I discussed above, the manuscript also includes looser language regarding biologic plausibility that is difficult to pin to concrete claims and supporting evidence. Whether the proposed model is more biologically plausible or represents a "biological constraint" is a difficult argument to make without substantial insight and evidence into the workings of the model and of human visual systems (for which many topics are not settled science). This issue was noted by Reviewer n3FM and was not clarified to their satisfaction even after a few rounds of revision. I tend to agree that these claims are not well established and would expect authors to make it clear that their model is constrained to behave in a way inspired by human visual processing behavior -- i.e. iterative glimpsing with foveated vision. Beyond that, I do not see evidence to claim that the proposed method is an apt model of human glimpsing or that the constraint is "biological" in a satisfying way. Perhaps biologically-inspired. Other reviewers did not raise this concern. I do not think this fundamentally changes the contributions of the paper, but should be addressed to avoid adding to confusion.

---

> ### Author Response · Authors · 2022-10-28
> **Camera ready version uploaded**
>
> We thank the AE for the extensive analysis which helped to better frame our work and contributions.
>
> We modified the paper accordingly, to refer to biologically inspired constraints (rather than biological constraints). These changes are already present in the title, and additional adjustments in all sections. We believe that this terminology helps the reader to better frame the actual contribution of our paper.
>
> We propose a new metric, String-Based Time-Delay Embeddings (SBTDE), which is only intuitively motivated. As well pointed out by the reviewer, no experiments are presented that show its superiority over the other metrics (that are anyway included in the article, too). Therefore, we reduced our claim to say that we present an “alternative” metric.
>
> NeVA-O (optimized version) is optimized with respect to the foveation step, and not over time. We find the optimal position for every individual fixation without considering future fixations.  In other words, we assume perferct information at every individual fixation but not perfect information with respect to the future. A version that optimizes with respect to the whole scanpath history would need a different (recurrent) formulation of the NeVA algorithm, therefore it is not analyzed in this paper.
>
> Practical use of the proposed methodology is justified in scenarios where its unfeasible to process a full resolution image. We think that the experiment described in section 4.4 shows the usefulness of NeVA in this specific case where resource constraints are applied. Examples of such applications are given in the discussion.